# Loss of the molecular clock in myeloid cells exacerbates T cell-mediated CNS autoimmune disease

Caroline E. Sutton[1], Conor M. Finlay[1], Mathilde Raverdeau[1], James O. Early[2,3], Joseph DeCourcey[1], Zbigniew Zaslona[2], Luke A.J. O'Neill[2], Kingston H.G. Mills [1] & Annie M. Curtis[3]

The transcription factor BMAL1 is a core component of the molecular clock, regulating biological pathways that drive 24 h (circadian) rhythms in behaviour and physiology. The molecular clock has a profound influence on innate immune function, and circadian disruption is linked with increased incidence of multiple sclerosis (MS). However, the mechanisms underlying this association are unknown. Here we show that BMAL1 and time-of-day regulate the accumulation and activation of various immune cells in a CNS autoimmune disease model, experimental autoimmune encephalomyelitis (EAE). In myeloid cells, BMAL1 maintains anti-inflammatory responses and reduces T cell polarization. Loss of myeloid BMAL1 or midday immunizations to induce EAE create an inflammatory environment in the CNS through expansion and infiltration of IL-1β-secreting $CD11b^+Ly6C^{hi}$ monocytes, resulting in increased pathogenic $IL-17^+/IFN-\gamma^+$ T cells. These findings demonstrate the importance of the molecular clock in modulating innate and adaptive immune crosstalk under autoimmune conditions.

[1] Immune Regulation Research Group, Trinity Biomedical Sciences Institute, Trinity College Dublin, 152-160 Pearse Street, D02 R590 Dublin, Ireland. [2] Inflammatory Research Group, School of Biochemistry and Immunology, Trinity Biomedical Sciences Institute, Trinity College Dublin, 152-160 Pearse Street, D02 R590 Dublin, Ireland. [3] Department of Molecular and Cellular Therapeutics, Royal College of Surgeons in Ireland (RCSI), 123 St. Stephens Green, D02 YN77 Dublin, Ireland. Kingston H.G. Mills and Annie M. Curtis contributed equally to this work. Correspondence and requests for materials should be addressed to K.H.G.M. (email: kingston.mills@tcd.ie) or to A.M.C. (email: anniecurtis@rcsi.com)

Life follows a 24 h rhythm driven by the daily cycles of light and dark due to the earth's rotation. The molecular clock is the timekeeping system within all our cells that integrates many aspects of our behaviour and physiology to align with these external rhythmic changes. The master clock resides in the suprachiasmatic nucleus (SCN) of the brain and promotes synchrony of rhythms throughout the body by signalling to peripheral clocks[1], such as in the liver[2], heart[2], muscle[3], immune system[4, 5], intestine[6] and even the microbiota[7].

The SCN clock keeps peripheral clocks in harmony via the hypothalamus pituitary adrenal axis and the autonomic nervous system through their respective hormones, glucocorticoids and catecholamines (epinephrine and norepinephrine). These hormones act as synchronizing messengers, or zeitgebers, to peripheral clocks[8, 9]. In addition to glucocorticoids and catecholamines, other hormones such as prolactin and growth hormone that are known to affect the immune system, also peak at certain times of the day. The control by the SCN on these autonomic and endocrine outputs keeps peripheral clocks, including that of immune cells, in phase with each other and allows for the coordination of a temporal programme of physiology across many tissues[10].

These peripheral clocks can also be influenced independently by cues such as fasting or feeding[11]. Coordination of these circadian rhythms relies on a number of transcriptional-translational feedback loops of core clock proteins. Most important amongst them is the basic helix–loop–helix PAS (bHLH-PAS) transcription factor BMAL1 (also known as ARNTL or MOP3), which forms a heterodimer with another bHLH-PAS transcription factor, appropriately named CLOCK (circadian locomotor output cycles kaput). The BMAL1:CLOCK heterodimer binds to E-box sequences on the genome and controls the transcriptional repressors Period and Cryptochrome. Inhibition in the dark phase of BMAL1:CLOCK by the nuclear accumulation of the PERIOD:CRYPTOCHROME complex allows for circadian oscillations in BMAL1:CLOCK activity on the gene promoters of thousands of downstream targets, classified as clock control genes (CCG). $Bmal1^{-/-}$ cells lack a functional molecular clock and all rhythms in clock gene expression and CCGs are ablated[12].

It has been established that a functional clock exists in macrophages[5, 13, 14] and that this clock has a major function in susceptibility to bacterial infection[15, 16], endotoxin challenge[17, 18] and cardiovascular disease[19]. Monocyte sub-populations are influenced by their intrinsic molecular clock such that the numbers of circulating CD11b$^+$ and Ly6C$^{hi}$ monocytes vary across the 24 h cycle[5, 16]. Loss of BMAL1 in the myeloid lineage promotes increased numbers and trafficking of the pro-inflammatory Ly6C$^{hi}$ monocytes into tissues and causes enhanced lethality upon *Listeria monocytogenes* infection[16]. Overall, loss of *Bmal1* in myeloid cells causes increased inflammatory responses[20], correlating with increased IL-1β and IFN-γ production[5, 16] and reduced expression of the anti-inflammatory cytokine IL-10[17].

For adaptive immunity, circadian oscillations of CCGs have been observed in T and B cells. Regulation of the adaptor protein ZAP70, which controls antigen-induced T cell proliferation, is regulated in a circadian manner, leading to T cell responses that are dependent on time-of-day[21]. Furthermore, there appears to be subset-specific requirements for clock genes in T helper cell development, with the loss of the clock component *Nfil3* (also known as *E4bp4*), leading to enhanced Th17 responses but not Treg cell development[22].

Intriguingly, there is an association between the circadian system and the immune-mediated demyelinating disorder, multiple sclerosis (MS). A large population-based study identified that teenagers conducting shiftwork, which causes circadian disruption, have an increased risk in the development of MS in later life[23]. Furthermore, incidences of relapse in MS occur most frequently in spring and summer, an observation attributed to lower levels of the clock-regulated hormone melatonin[24].

There are conflicting reports on the expression of *Bmal1* in T cells and function of Bmal1 in the development of experimental autoimmune encephalomyelitis (EAE), a murine model for MS. Hemmers et al.[25] showed that there is no effect on development of disease in T cell-specific *Bmal1* knockout mice, but Druzd et al.[26], in a more comprehensive analysis, reported that loss of *Bmal1* in T cells affects the severity of EAE. In addition to T cells, myeloid lineage cells also have a pathogenic function in EAE[27, 28]. Myeloid cells migrate across the blood–brain barrier during EAE[29] and secrete IL-1[30, 31] and granulocyte-macrophage colony-stimulating factor (GM-CSF)[32] to modulate the development of EAE.

Therefore, we hypothesized that BMAL1 expression and the molecular clock in myeloid cells might be important in CNS autoimmune disease through modulation of innate immunity. Here we show that mice lacking myeloid *Bmal1* and mice immunized at midday develop enhanced EAE diseases through expansion and infiltration of IL-1β-secreting CD11b$^+$Ly6C$^{hi}$ monocytes into the CNS. Our results provide new opportunities to enhance circadian function or time-of-day drug-targeting strategies to alleviate autoimmune disease.

## Results

**Loss of myeloid *Bmal1* induces pro-inflammatory cytokines.** Loss of *Bmal1* from Lyz2 lineage cells, which include monocytes, has been shown to enhance the numbers of Ly6C$^+$ monocytes and production of the pro-inflammatory cytokines IL-1β and IFN-γ, leading to increased lethality in a model of septic shock[16]. We found that the Th1-polarizing cytokine IL-12p40 was significantly enhanced under basal conditions in the serum of $Bmal1^{LoxP/LoxP}Lyz2^{Cre}$ ($Bmal1^{Myeloid-/-}$) mice when compared with control $Lyz2^{Cre}$ ($Bmal1^{Myeloid+/+}$) mice (Fig. 1a) ($p < 0.001$, Mann–Whitney U test).

We next examined GM-CSF or M-CSF-induced differentiation of bone marrow (BM)-derived cells from $Bmal1^{Myeloid-/-}$ mice and $Bmal1^{Myeloid+/+}$ controls. While GM-CSF stimulation of BM cells expanded a CD11b$^+$CD11c$^{hi}$MHCII$^{hi}$ population, M-CSF stimulation increased the frequency of CD11b$^+$CD11c$^{lo/int}$F4/80$^+$ cells (Supplementary Fig. 1a). We confirmed that GM-CSF and M-CSF expanded cells from the bone marrow of $Bmal1^{Myeloid-/-}$ were indeed devoid of *Bmal1* (Supplementary Fig. 1b, c). Interestingly, CD11c$^+$ cells isolated from the spleen of $Bmal1^{Myeloid-/-}$ mice did not appear to be of the Lyz2 lineage; these cells displayed similar levels of *Bmal1* in comparison with cells isolated from $Bmal1^{Myeloid+/+}$ mice (Supplementary Fig. 1d).

We found that GM-CSF-expanded BM cells produced higher concentrations of IL-1β, IL-23, but not IL-10 in response to *Mycobacterium tuberculosis* (MTB), when compared with cells from $Bmal1^{Myeloid+/+}$ controls (Fig. 1b) ($p < 0.01$, Mann–Whitney U test), however no significant increase in these cytokines was observed in MTB-stimulated M-CSF-expanded BM from $Bmal1^{Myeloid-/-}$ mice (Supplementary Fig. 2). Surface expression of MHCII was higher on GM-CSF-expanded BM cells from $Bmal1^{Myeloid-/-}$ mice (Fig. 1c) ($p < 0.01$, Mann–Whitney U test). Interestingly, not only were inflammatory pathways enhanced in the absence of *Bmal1* in CD11b$^+$ cells, there was also a loss of the immune checkpoint inhibitors PD-L1 ($p < 0.05$, Mann–Whitney U test) but not PD-L2, on MHCII$^+$CD11b$^+$CD11c$^+$ cells from $Bmal1^{Myeloid-/-}$ compared with $Bmal1^{Myeloid+/+}$ mice (Fig. 1d). Given that the loss of *Bmal1* from GM-CSF-expanded BM cells significantly increased the expression of IL-12p40, IL-1β and IL-23, cytokines associated

with the polarization of T helper cells into either Th1 or Th17 cells, we next examined whether loss of *Bmal1* from Lyz2 lineage cells would result in expansion of these T cell subsets. Co-culture of GM-CSF-expanded BM cells with MOG-specific CD4 T cells, isolated from *Bmal1*$^{Myeloid+/+}$ mice 7 d after immunization with MOG + CFA, produced higher concentrations of IFN-γ when the BM was derived from *Bmal1*$^{Myeloid-/-}$ mice (Fig. 1e) ($p < 0.05$,

Mann–Whitney $U$ test). In contrast, there was no significant difference in antigen-specific T cell responses after co-culture with M-CSF-expanded BM cells from *Bmal1*$^{Myeloid-/-}$ mice compared to *Bmal1*$^{Myeloid+/+}$ mice (Supplementary Fig. 3) ($p < 0.05$, Mann–Whitney $U$ test).

Furthermore, MOG-specific CD4 T cells taken from *Bmal1*$^{Myeloid-/-}$ mice, which are themselves not of the Lyz2

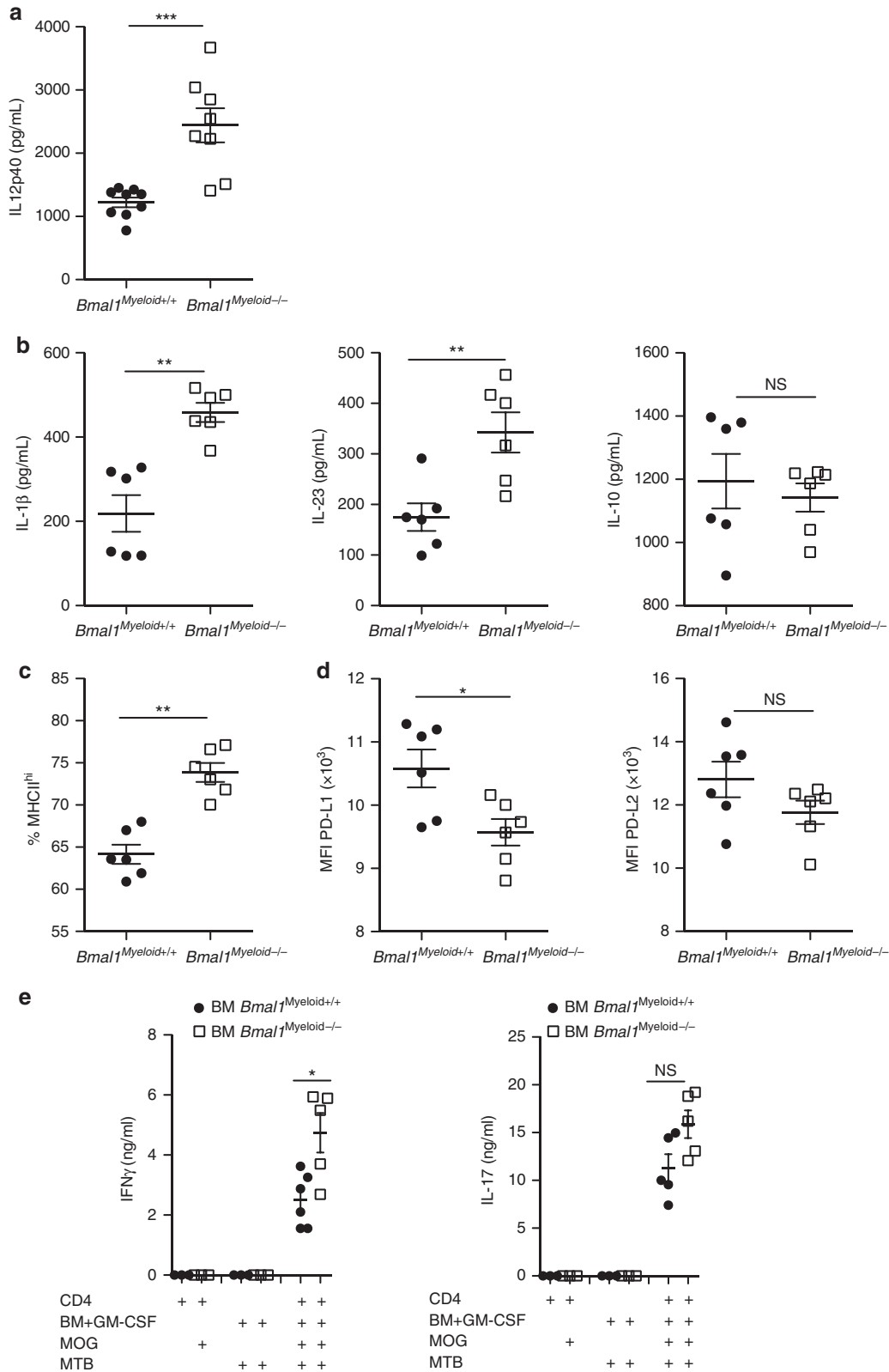

lineage and therefore express $Bmal1$, and co-cultured with MOG and MTB-stimulated GM-CSF-expanded BM from $Bmal1^{Myeloid+/+}$ mice, produced significantly more antigen-specific IL-17 and IFN-γ when compared with CD4 T cells isolated from MOG-immunized $Bmal1^{Myeloid+/+}$ mice (Supplementary Fig. 4, $p < 0.01$, Mann–Whitney $U$ test). These results indicate that the absence of $Bmal1$ from myeloid cells results in a hyper-inflammatory environment leading to enhanced Th1 and Th17 responses.

**Myeloid $Bmal1$ attenuates the development of EAE.** We next examined whether loss of $Bmal1$ from myeloid lineage cells influenced the development of EAE, where Th1 and Th17 as well as Ly6C[+] cells have been implicated in the development of disease[29, 31]. Immunization of $Bmal1^{Myeloid-/-}$ and $Bmal1^{Myeloid+/+}$ mice with MOG + CFA and low dose of pertussis toxin (PT) led to the development of EAE in these mice (Fig. 2a). The lower dose PT led to less severe disease, however, mice lacking $Bmal1$ from myeloid cells had a significantly more severe course of disease than $Bmal1^{Myeloid+/+}$ mice. Low dose PT was used in all subsequent experiments between $Bmal1^{Myeloid-/-}$ and $Bmal1^{Myeloid+/+}$ mice. The increased severity of EAE in the $Bmal1^{Myeloid-/-}$ mice correlated with an increased pro-inflammatory environment in the CNS of these mice. We found significantly increased mRNA expression of $Il1b$, $Ifng$, $Ccl2$ ($p < 0.01$, Mann–Whitney $U$ test) and $Il17$ ($p < 0.05$, Mann–Whitney $U$ test) in the brains of $Bmal1^{Myeloid-/-}$ mice with EAE when compared with controls (Fig. 2b). Dysregulation of pro-inflammatory responses in the $Bmal1^{Myeloid-/-}$ mice occurred rapidly after the induction of EAE. We found enhanced numbers of CD11b[+]Ly6C[hi] cells in the draining lymph nodes (LN) 3 days post immunization with MOG and CFA (Fig. 2c) ($p < 0.05$, Mann–Whitney $U$ test), and significantly increased numbers of MHCII[+] cells in these mice compared with controls (Fig. 2d) ($p < 0.001$, Mann–Whitney $U$ test).

We next isolated infiltrating mononuclear cells from the spinal cords on d 10 post immunization with MOG + CFA and low dose PT and examined whether loss of $Bmal1$ from the myeloid compartment affected the cellular infiltrate into the CNS. We found significantly more CNS infiltrating immune cells in $Bmal1^{Myeloid-/-}$ mice compared with $Bmal1^{Myeloid+/+}$ mice. There were increases in the numbers and percentages of CD45[+] cells (Fig. 3a) as well as CD11b[+]Ly6C[hi] cells (Fig. 3b). We found that the Ly6C[hi]CCR2[+] cells, which infiltrate the brain during EAE, were entirely CD11b[+] (Supplementary Fig. 5a, b), indicating that CD11b[+]Ly6C[hi]CCR2[+] cells are important for the development of EAE.

The frequency and absolute numbers of CD11b[+]Ly6C[hi] cells was enhanced in the CNS of $Bmal1^{Myeloid-/-}$ compared with control mice (Fig. 3b). This population has been identified as one of the key cell populations crossing the blood–brain barrier and producing IL-1β and mediating neuroinflammation leading to the development of EAE[29]. Indeed we found enhanced numbers of CD11b[+]Ly6C[hi] cells producing IL-1β in the spinal cord (Fig. 3c)

($p < 0.01$, Mann–Whitney $U$ test) and brain (Supplementary Fig. 5d) of $Bmal1^{Myeloid-/-}$ mice with EAE compared with controls. Furthermore, there was an increase in the percentage of IL-1β production by these CD11b[+]Ly6C[hi] cells, indicating that not only are there enhanced numbers of these cells entering the CNS of $Bmal1^{Myeloid-/-}$ mice, but that they produce significantly more IL-1β in the CNS, compared to the same cells which enter the CNS of $Bmal1^{Mye+/+}$ mice. In addition, the CD11b[+]Ly6C[hi] population is responsible for significant IL-1β production in the CNS of mice with EAE (Fig. 3d) ($p < 0.05$, Mann–Whitney $U$ test).

The increased infiltration of inflammatory populations of cells into the brains and enhanced severity of EAE in $Bmal1^{Myeloid-/-}$ mice correlated with enhanced MOG-specific T cell responses. LN cells from mice on d 14 of EAE secreted GM-CSF, IFN-γ and IL-17, but not IL-10, in response to re-stimulation with MOG in vitro (Fig. 4a). The concentrations of these cytokines was significantly higher in cells derived from $Bmal1^{Myeloid-/-}$ mice compared with $Bmal1^{Myeloid+/+}$ mice ($p < 0.05$ and $p < 0.001$, Mann–Whitney $U$ test). Flow cytometry analysis revealed that there were significantly increased numbers of CD3 T cells, where $Bmal1$ was depleted, particularly CD4 T cells entering the CNS of $Bmal1^{Myeloid-/-}$ mice compared with controls (Fig. 4b–d) ($p < 0.05$, Mann–Whitney $U$ test). Consistent with these data, we demonstrated significantly enhanced infiltration of IL-17[+] and IFN-γ[+] cells into the brains of $Bmal1^{Myeloid-/-}$ mice compared to controls (Fig. 4e) ($p < 0.01$ and $p < 0.05$, Mann–Whitney $U$ test). Collectively our findings demonstrate that loss of $Bmal1$ in myeloid cells is associated with enhanced innate and adaptive immune responses and a more severe course of CNS autoimmune disease.

**$Bmal1$ and $Reverb\alpha$ inversely correlates with inflammation.** We compared spinal cords from mice with severe EAE with naive control mice and found significantly less $Bmal1$ and $Reverb\alpha$ (a target gene of BMAL1 and core clock component) mRNA expression in the mice with severe EAE (Fig. 5a) ($p < 0.01$, Mann–Whitney $U$ test). This dysregulation was specific for these two core clock components as the levels of $Clock$ and $Per2$ were not significantly altered in mice with EAE when compared with naive control mice, demonstrating that the circadian feedback loops are disrupted during development of EAE. When mice were immunized with CFA and PT, without MOG, we did not observe this decrease expression of $Bmal1$ and $Reverb\alpha$ mRNA in the spinal cord, indicating that the effects on clock gene expression are specific to EAE and not due to systemic inflammation (Supplementary Fig. 6). The decrease in $Bmal1$ and $Reverb\alpha$ mRNA expression correlated with increases in $Ccl2$, $Il1b$ and $Csf2$ gene expression in the spinal cord of mice with EAE (Fig. 5b) ($p < 0.01$ and $p < 0.05$, Mann–Whitney $U$ test). Crucially, we also found that loss of $Bmal1$ expression in the spinal cord significantly correlated with an increase in the severity of EAE (Fig. 5c) ($p < 0.001$, linear regression).

**Fig. 1** Loss of myeloid $Bmal1$ increases pro-inflammatory responses. **a** Sera from naive $Bmal1^{Myeloid+/+}$ and $Bmal1^{Myeloid-/-}$ mice were tested by ELISA for IL-12p40 ($n = 7$–9). **b** Bone marrow (BM) cells isolated from $Bmal1^{Myeloid+/+}$ or $Bmal1^{Myeloid-/-}$ mice was cultured with granulocyte-macrophage colony-stimulating factor (GM-CSF) (20 ng/ml). After 6 days cells were harvested and stimulated with $Mycobacterium\ tuberculosis$ (MTB) (100 μg/ml). Supernatants were removed at 24 h and tested by ELISA for cytokine production. **c**, **d** Day 6 GM-CSF-expanded BM cells from $Bmal1^{Myeloid+/+}$ and $Bmal1^{Myeloid-/-}$ mice were stained for MHCII, PD-L1 or PD-L2 gating on CD11b[+]CD11c[+] cells ($n = 6$). **e** BM cells from either $Bmal1^{Myeloid+/+}$ and $Bmal1^{Myeloid-/-}$ mice was grown in the presence of GM-CSF. After 6 days cells were harvested and incubated with MTB (100 μg/ml) or myelin oligodendrocyte glycoprotein (MOG$_{35-55}$) (25 μg/ml) + MTB for 3 h prior to the addition of MACS purified CD4[+] T cells isolated from draining lymph node (LN) of 7 d MOG + complete Freund's adjuvant (CFA) immunized $Bmal1^{Myeloid+/+}$ mice. After 72 h of co-culture supernatants were removed and IFN-γ and IL-17 concentrations determined by ELISA ($n = 5$–6). Statistics were performed by Mann–Whitney $U$ test **b**–**e**. All data presented as means±standard error of the mean (SEM). *$p < 0.05$; **$p < 0.01$; ***$p < 0.001$

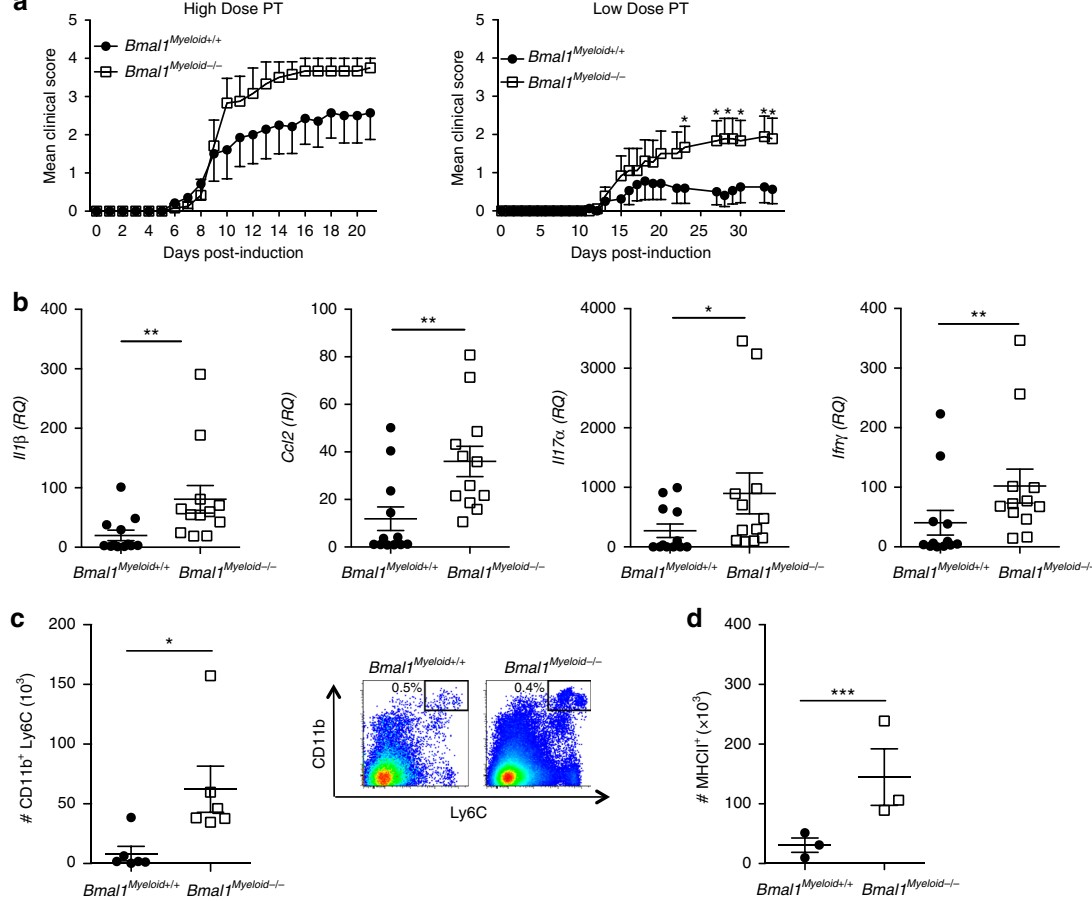

**Fig. 2** Loss of *Bmal1* from myeloid cells exacerbates EAE. **a** *Bmal1*<sup>*Myeloid+/+*</sup> or *Bmal1*<sup>*Myeloid−/−*</sup> mice were immunized to develop experimental autoimmune encephalomyelitis (EAE) with myelin oligodendrocyte glycoprotein (MOG$_{35-55}$) + complete Freund's adjuvant (CFA) on d 0, and with either low dose pertussis toxin (PT) (125 ng/mouse) or high dose PT (250 ng/mouse) on d 0 and d 2. Mice were graded for EAE daily. Statistics by Kruskal–Wallis with at least 6 mice per group. **b** RT-PCR analysis of whole brain d 10 post induction of EAE examining *Il1b*, *Ccl2*, *Il17a* and *Ifng* expression (n = 9–12). **c**, **d** D 3 post immunization draining lymph nodes (LN) were isolated from *Bmal1*<sup>*Myeloid+/+*</sup> or *Bmal1*<sup>*Myeloid−/−*</sup> mice and stained ex vivo for CD3, CD11b, Ly6C and MHCII, gating on live CD3⁻ cells (n = 3–6). Statistics were performed by Mann–Whitney U test (**b**–**d**). All data presented as means± standard error of the mean (SEM). RQ relative quantification. *p < 0.05; **p < 0.01; ***p < 0.001

**Effect of time-of-day immunization on severity of EAE**. In the mouse facility, Zeitgeber time is a measure of time (hours) after lights go on; ZT0 corresponds to Zeitgeber time 0 and lights on whereas ZT12 corresponds to Zeitgeber time 12, lights off. We found that LN cells taken from mice at ZT6 produced higher concentrations of IL-1β when compared with LN cells taken at ZT18 after stimulation for 24 h with MTB or medium (Fig. 6a) (p < 0.05, Mann–Whitney U test). There was no difference in MTB-induced TNF at ZT6 compared to ZT18. This correlated with diurnal oscillations in the numbers of CD11b⁺Ly6C<sup>hi</sup> cells in the spleen peaking at ZT6 (Supplementary Fig. 7a), no diurnal variation was observed for CD11c⁺MHCII⁺ although it was previously shown that gene expression in CD11c⁺MHCII⁺ and macrophage populations peaks at ZT6[13].

We examined whether the time-of-day at point of immunization with MOG and CFA would affect the course of EAE. We found that the course of disease was significantly more severe in wild-type mice immunized at ZT6 (Fig. 6b) (p < 0.05, Kruskal–Wallis test and Mann–Whitney test of area under the curve), equating to middle of daylight hours, compared with mice immunized at ZT18. We confirmed this trend, using mice that had the light/dark schedule inverted using a light cabinet, in which ZT6 and ZT18 mice and ZT0 and ZT12 mice were

immunized at the same solar time to minimize any experimental error associated with the preparation or storage of the MOG and CFA emulsion (Supplementary Fig. 7b). We then induced EAE in *Bmal1*<sup>*Myeloid+/+*</sup> vs. *Bmal1*<sup>*Myeloid−/−*</sup> at ZT6 and ZT18. Consistent with our earlier data, we observed more severe disease in *Bmal1*<sup>*Myeloid+/+*</sup> mice immunized at ZT6 compared with ZT18 (Fig. 6c). Strikingly, mice lacking *Bmal1* in myeloid cells no longer displayed this time-of-day difference, with a similar clinical course of EAE when induced at ZT6 and Z18 (Fig. 6c) (p < 0.05, Kruskal–Wallis U test and Mann–Whitney test of area under the curve). Taken together, these results clearly demonstrate an impact of clock time on induction of disease in the EAE model, even though clinical signs take a week or longer to appear, and that the myeloid clock is responsible for this time-of-day effect.

We next examined the peritoneal exudate cells (PEC) from mice immunized either at ZT6 or ZT18 to determine if time of immunization influenced innate immune cell accumulation and IL-1β production. We found that mice immunized at ZT6 had higher numbers and percentages of CD11b⁺ populations, including Ly6C<sup>hi</sup> and Ly6G⁺, compared to both naive and ZT18-immunized mice (Fig. 6d). Furthermore, there were significantly more IL-1β-producing Ly6G⁺ cells when mice were

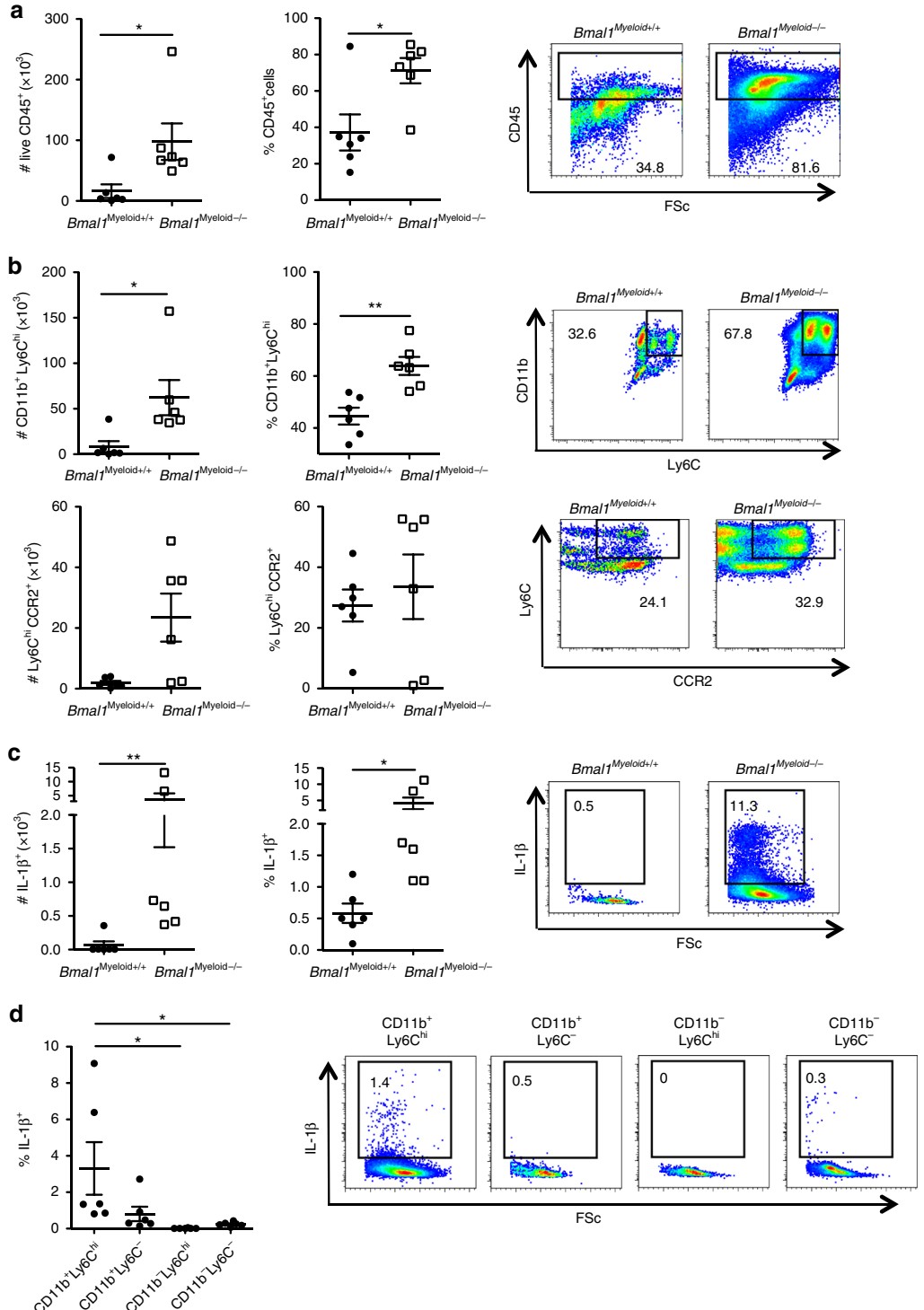

**Fig. 3** Enhanced accumulation of inflammatory CD11c⁺Ly6C⁺ cells producing IL-1β in the CNS of *Bmal1*^Myeloid−/− mice during EAE. **a–c** *Bmal1*^Myeloid+/+ and *Bmal1*^Myeloid−/− mice were immunized to develop experimental autoimmune encephalomyelitis (EAE) with myelin oligodendrocyte glycoprotein (MOG$_{35-55}$) + complete Freund's adjuvant (CFA) on d 0 and with pertussis toxin (PT) (125 ng/mouse) on d 0 and d 2 (*n* = 6). **a** D 10 post EAE induction live cells infiltrating into the spinal cord were FACS stained for CD45 (*n* = 6). **b** Live CD45⁺ cells were examined for Ly6C v CD11b or CCR2 v Ly6C (*n* = 6). **c** IL-1β expression was determined in live CD45⁺CD11b⁺Ly6C⁺ cells (*n* = 6). **d** 10 d post immunization CD11b and Ly6C populations infiltrating the spinal cord of *Bmal1*^Myeloid−/− mice were examined for IL-1β expression by FACS, gating on live CD45⁺ cells (*n* = 6). Data as means ±standard error of the mean (SEM). Statistics were performed by Mann–Whitney *U* test **a**–**d**. *$p < 0.05$; **$p < 0.01$

immunized at ZT6 compared with ZT18 (Fig. 6e) ($p < 0.05$, Mann–Whitney *U* test). The number of IL-1β-producing Ly6C⁺ cells were also increased, though not significantly, in mice immunized at ZT6 compared with ZT18 (Fig. 6e). These findings

indicate that the increase in disease severity after immunization at ZT6 compared with ZT18 can be attributed to enhanced accumulation of inflammatory monocytes and increased IL-1β production, similar to that observed in *Bmal1*^Myeloid−/− mice.

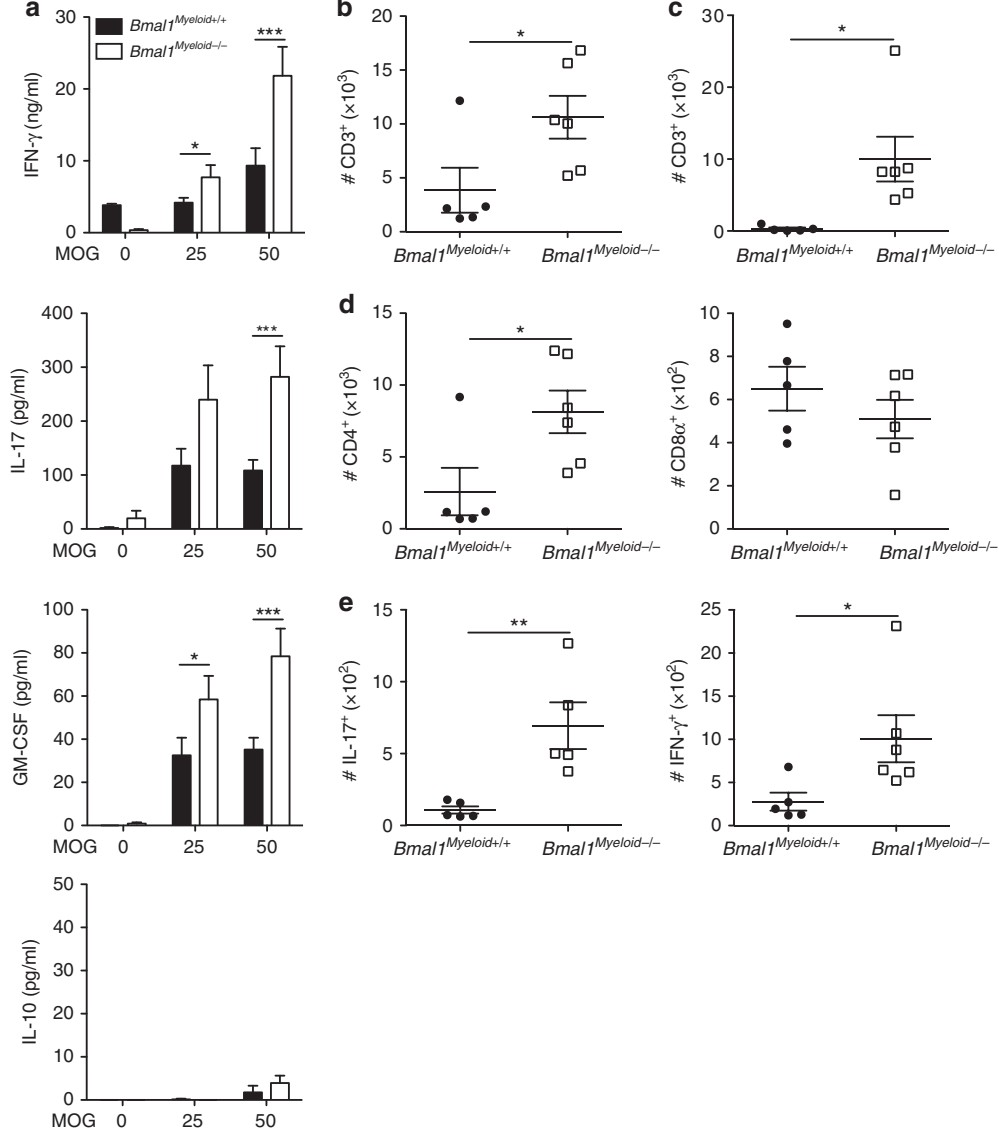

**Fig. 4** Enhanced Th1 and Th17 responses in the CNS of $Bmal1^{Mye-/-}$ mice with EAE. **a–e** $Bmal1^{Myeloid+/+}$ or $Bmal1^{Myeloid-/-}$ mice were immunized with myelin oligodendrocyte glycoprotein (MOG$_{35-55}$) + complete Freund's adjuvant (CFA) on d 0 and with pertussis toxin (PT) (125 ng/mouse) on d 0 and d 2. **a** After 14 d draining lymph node (LN) cells were re-stimulated in the presence and absence of MOG$_{35-55}$ antigen (25 and 50 μg/ml) or with medium for 72 h. Supernatants were removed and tested for IL-17, GM-CSF, IFN-γ and IL-10. Statistics by one way ANOVA with Tukey's post-test of triplicate assay of 6 mice per group. **b–e** 10 d post immunization brain and spinal cords were isolated and surface stained for CD45, CD3, CD8α and CD4 and intracellularly for IL-17 and IFN-γ. **b** Cells were gated on live CD45$^+$ cells in the brain ($n=5–6$). **c** Cells were gated on live CD45$^+$ cells spinal cord ($n=5–6$). **d** Cells were gated on live CD45$^+$CD3$^+$ cells in the brain ($n=5–6$). **e** Cells were gated on live CD45$^+$ cells in the brain ($n=5–6$). Statistics were performed by Mann–Whitney $U$ test (**b–e**). All data presented as means±standard error of the mean (SEM). *$p < 0.05$; **$p < 0.01$; ***$p < 0.001$

## Discussion

The significant finding of this study is that expression of the clock gene *Bmal1* and time-of-day in myeloid cells regulates the innate and adaptive immune responses that mediate autoimmune diseases. We demonstrate that mice with a targeted deletion of *Bmal1* from myeloid cells have significantly enhanced pro-inflammatory and T cell-polarizing cytokine responses, resulting in heightened inflammation and enhanced susceptibility to the induction of autoimmune disease.

There is growing evidence of the influence of the molecular clock and time-of-day on immune function. Shift workers, frequent air travellers and populations at greater risk of circadian disruption have an increased incidence of chronic inflammatory disease[33]. Furthermore, animal models of jet-lag display an increase in the levels of pro-inflammatory cytokines, in particular

IL-1β[34] and in mice the circadian clock directly controls inflammatory arthritis[35]. Patients with rheumatoid arthritis display a peak in disease severity in the early morning as a result of an enhanced pro-inflammatory cytokine production at this time, which may reflect lower levels of clock-regulated cortisol[36]. Modulation of cellular homing[26, 37] or clock control of toll-like receptor expression[38] provide alternative explanations for this time-of-day effect. There is also emerging evidence that the macrophage molecular clock has profound influence on epigenetic control of the inflammatory response[39].

It has been reported that deletion of the core clock gene *Bmal1* from cells of the Lyz2 lineage causes an expansion of Ly6C$^{hi}$ monocytes, and is associated with increase inflammatory cytokine production and enhanced lethality in a sepsis model[16]. These findings suggest an anti-inflammatory role for *Bmal1* in this cell

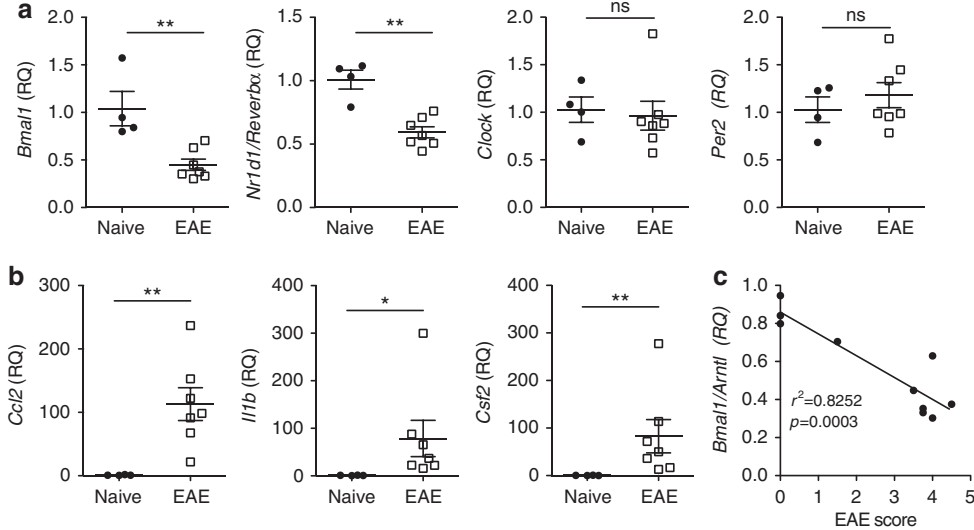

**Fig. 5** Dysregulation of *Bmal1* and *Reverba* during the acute disease phase of EAE. **a–c** Spinal cords were removed from 14 d myelin oligodendrocyte glycoprotein (MOG$_{35-55}$), complete Freund's adjuvant (CFA) and pertussis toxin (PT)-immunized or naive female C57BL/6 mice. **a, b** RT-PCR analysis of isolated spinal cords ($n = 4$–6). **c** Linear regression of clinical scores of 10 mice vs. *Bmal1* gene expression in the same animals ($n = 10$). Statistics were performed by Mann–Whitney $U$ test (**a–c**). All data presented as means±standard error of the mean (SEM). RQ relative quantification.*$p < 0.05$; **$p < 0.01$

population. Furthermore, we have previously shown that lipopolysaccharide (LPS)-activation of macrophages suppressed *Bmal1* mRNA expression via the microRNA MiR-155[17], demonstrating that noxious stimuli may inhibit the molecular clock to promote a pro-inflammatory environment.

Consistent with previous reports that loss of *Bmal1* from myeloid cells causes a hyper-inflammatory environment[16, 19], we found significantly increased production of the pro-inflammatory Th1-polarizing cytokine IL-12p40 in the serum of mice lacking myeloid *Bmal1*. In addition, loss of *Bmal1* from GM-CSF-expanded BM cells significantly enhanced MHC class II expression along with IL-1β and IL-23 in response to MTB. Consistent with the enhanced MHC class II expression on T cells and production of T cell-polarizing cytokines by myeloid cells, we found that the GM-CSF-expanded BM cells from mice that lacked *Bmal1*-induced significantly more IFN-γ production by MOG-specific T cells. While loss of *Bmal1* enhanced pro-inflammatory responses, it did not affect production of the anti-inflammatory cytokine IL-10, although reduced expression of IL-10 has been observed in M-CSF expanded BM cells from *Bmal1*$^{Myeloid-/-}$ in response to LPS[17], indicating that the clock is modulating responses that are both cell and stimulation specific. Furthermore, expression of the immune checkpoint protein PD-L1 was significantly decreased on CD11b$^+$CD11c$^+$ cells from *Bmal1*$^{Myeloid-/-}$ mice.

The neuroinflammation observed in mice with EAE is thought to be mediated by adaptive immune cells, especially autoantigen-specific Th1 and Th17 that contribute to demyelination[31, 40]. However, there have been conflicting reports on the role for *Bmal1* expression in CD4 T cells and the development of EAE[31, 40]. It has been shown that *Bmal1* expression in CD4 T cells can influence the development of EAE only if immunization occurs at a time-of-day corresponding to high numbers of both innate and adaptive cells residing in LNs[26]. However, EAE was attenuated in mice which had T cells lacking *Bmal1*, indicating that *Bmal1* is specifically protective in the myeloid lineage for autoimmunity. Other cell subsets, such as innate immune cells, have also been shown to be crucial for the development of EAE, especially inflammatory monocytes and neutrophils that home to the CNS and produce IL-1β[29]. IL-1β plays a critical role in neuroinflammation through synergy with IL-23 to promote expansion of

Th17 and IL-17-secreting γδ T cells[41]. Furthermore, it has been shown that there is higher IL-1β expressed in peritoneal macrophages isolated at ZT6 compared to ZT18[13]. We demonstrate that the severity of EAE was enhanced in mice lacking *Bmal1* from myeloid cells. The increase in disease severity correlated with significant infiltration of inflammatory monocytes, especially Ly6C$^{hi}$CD11b$^+$ cells which produce IL-1β during the course of EAE. Furthermore, we found that the targeted loss of *Bmal1* from myeloid cells resulted in enhanced MOG-specific Th1 and Th17 responses and increased IL-17, IFN-γ and GM-CSF-producing T cells in the CNS of mice with EAE.

Our findings suggest that there is dysregulation of clock genes during the inflammation that causes autoimmune disease of the CNS. Indeed we show the selective reduction in mRNA expression of *Bmal1* and *Reverba*, but not *Period2* or *Cryptochrome2*, in the CNS of mice with EAE. At the height of EAE, the loss of *Bmal1* was associated with significantly enhanced IL-17, IFN-γ and IL-1β production in the CNS. Autoimmune and degenerative/inflammatory disease such as rheumatoid arthritis and osteoarthritis can impact directly on molecular clock expression, thus potentially driving to a vicious cycle of inflammation[42]. Furthermore, we found diurnal oscillations in the number of inflammatory cells of the innate and adaptive immune system, suggesting that time-of-day can determine the magnitude of the immune response, consistent with a previous published report[26]. Indeed, therapeutic strategies which incorporate alignment to circadian rhythms, such as alignment of prednisolone administration to the endogenous circadian rhythm of cortisol, has proved efficacious in treating rheumatoid arthritis[43]. Vaccine-induced immune responses are also dependent on the clock; sleep after vaccination enhanced the Th1 responses induced by a hepatitis A vaccine[44, 45]. It appears that the adaptive immune response and induction of immunological memory is enhanced during the sleep phase. Another vaccine study involving TLR9 as the adjuvant reported that the time of vaccination influenced the adaptive immune responses assessed weeks later[38]. We found that immunization of mice with MOG and CFA during daylight hours and the rest period of the mouse, induced more robust EAE when compared to immunization of mice at night, with the effect on the course of disease manifesting weeks after immunization. The enhanced severity of EAE in mice immunized at ZT6 correlated

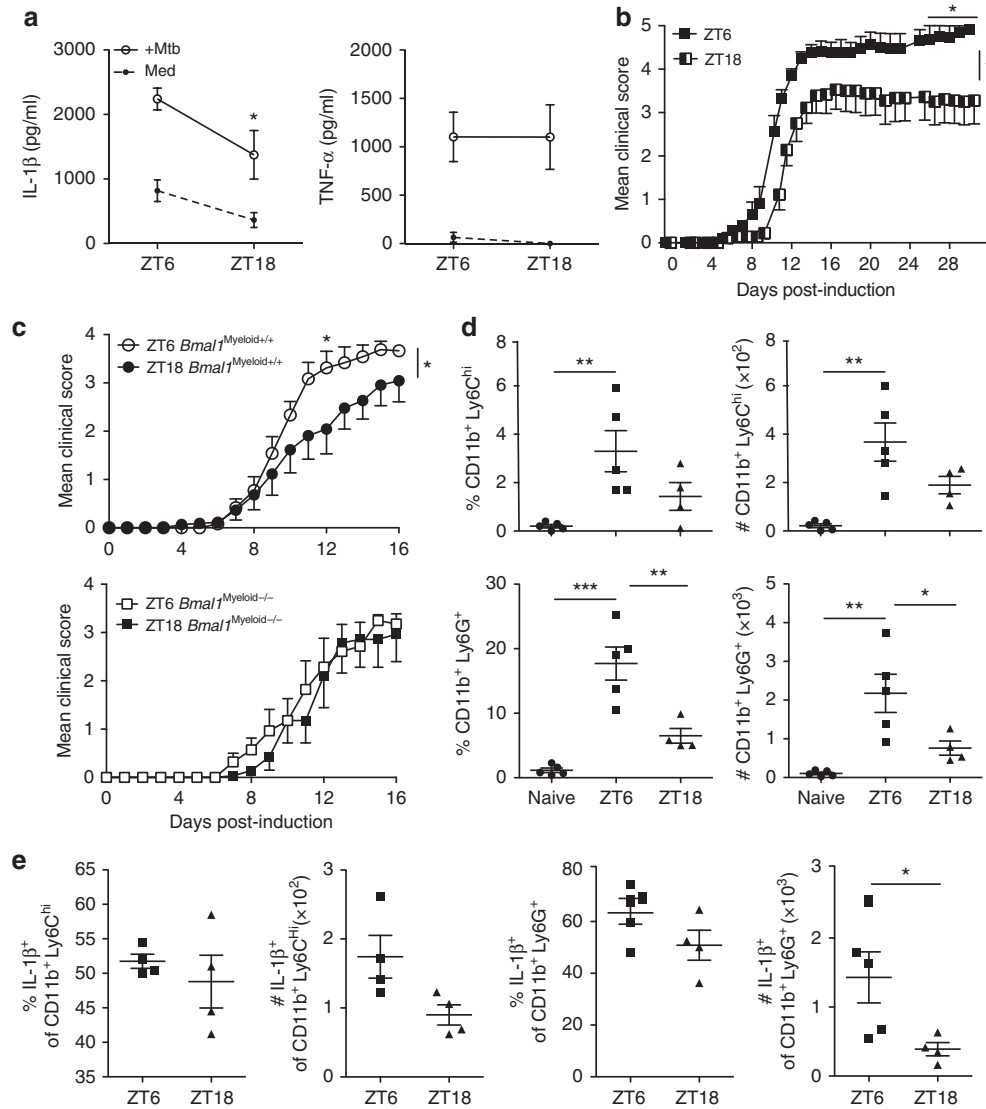

**Fig. 6** Severity of EAE is dependent upon time-of-day at point of immunization. **a** Lymph node (LN) cells isolated at either ZT6 or 18 were stimulated with medium or *Mycobacterium tuberculosis* (MTB) (100 μg/ml). Supernatants were tested by ELISA for IL-1β and TNF after 24 h. **b** Daily clinical score for myelin oligodendrocyte glycoprotein (MOG$_{35-55}$) + complete Freund's adjuvant (CFA) + pertussis toxin (PT)-induced EAE in female C57BL/6 mice, immunized at either ZT6 or ZT18. Statistics by Kruskal–Wallis test on indicated days of 12 mice per group and Mann–Whitney test of area under the curve. **c** EAE clinical scores in *Bmal1*$^{Myeloid+/+}$ and *Bmal1*$^{Myeloid-/-}$ female mice immunized with MOG + CFA and PT at either ZT6 or ZT18. Statistics by Kruskal–Wallis test on indicated days for 7–12 mice per group and Mann–Whitney test of area under the curve. **d**, **e** Peritoneal exudate cells were removed from 3 d MOG$_{35-55}$ + CFA + PT ZT6 or ZT18-immunized or naive female C57BL/6 mice (*n* = 4–5). **d** FACS was performed directly ex vivo gating on CD11b$^+$ cells and examining Ly6C$^{hi}$ and Ly6G$^+$ populations (*n* = 4–5). **e** Intracellular FACS for IL-1β production gating on CD11b$^+$Ly6C$^{hi}$ or CD11b$^+$Ly6G$^+$ populations (*n* = 4–5). Statistics were performed by Mann–Whitney *U* test (**a**, **d**, **e**). All data is presented as means±standard error of the mean (SEM). *$p < 0.05$; **$p < 0.01$; ***$p < 0.001$

with a hyper-inflammatory response, including enhanced numbers of inflammatory monocytes and IL-1β production. Significantly, we found that this time-of-day rhythmicity was dependent on the myeloid clock as the time-of-day response was lost in animals lacking myeloid *Bmal1*. Druzd et al.[26] also demonstrated that the time-of-day response was ablated with *Bmal1* depletion in CD4 T cells, indicating that both the myeloid and lymphoid clock influences this response.

Collectively, our findings demonstrate that the myeloid molecular clock and circadian rhythms can influence the development of autoimmune disease. *Bmal1* expression and a functional molecular clock in myeloid lineage cells appears to regulate these temporal variations, yielding a less pro-inflammatory environment overall. Our data provide mechanistic insights into how time-of-day and clock disruption in myeloid cells impacts on

autoimmunity thus providing opportunities to enhance circadian function or time-of-day drug-targeting strategies to alleviate autoimmune disease.

## Methods

**Mice.** All mice were maintained according to European Union regulations and the Irish Health Products Regulatory Authority. Experiments were performed under Health Products Regulatory Authority license with approval from the Trinity College Dublin BioResources Ethics Committee. Female C57BL/6 wild-type mice were bred in house from established colonies. All mice were housed under specific pathogen-free conditions and euthanised humanely by carbon dioxide.

**Gene-modified animals.** Mice with the gene *Bmal1*-containing LoxP sites at either side of exon 8 were obtained from Jackson Labs (stock no. 007668). *Bmal1*$^{LoxP/LoxP}$ were crossed with *Lyz2*$^{Cre}$, which express Cre recombinase under the control of the *Lyz2* promoter (which encodes LysozymeM) to produce progeny that have *Bmal1*

excised in the myeloid lineage (monocytes, macrophages and granulocytes). Female $Bmal1^{LoxP/LoxP}Lyz2^{Cre}$ ($Bmal1^{Myeloid-/-}$) mice where compared with control female $Lyz2^{Cre}$ ($Bmal1^{Myeloid+/+}$). Offspring were genotyped to confirm the presence of LoxP sites and Cre recombinase.

**Induction and assessment of EAE**. EAE was induced in 6–8 week old female C57BL/6, $Bmal1^{Myeloid-/-}$ and $Bmal1^{Myeloid+/+}$ mice by subcutaneous (s.c.) immunization with $MOG_{35-55}$ peptide (150 μg/mouse; Genscript) emulsified in complete Freund's adjuvant (CFA; Condrex) containing 4 mg/ml heat-killed MTB. Mice were injected intraperitoneally (i.p.) with 200 ng/mouse pertussis toxin (PT; Kaketsuken), unless otherwise stated, on d0 and d2. No PT was given to mice where draining LN were taken on d 3 post immunization. For the "time-of-day" EAE immunization, 6–8 week-old female C57BL/6 mice were immunized at either ZT0 (8.00 a.m.), ZT6 (2.00 p.m.), ZT12 (8.00 p.m.) or ZT18 (2.00 a.m.) or their time-of-day was adjusted using a light cabinet so that ZT0 and ZT12 both corresponded to 9am and ZT6 and ZT18 both corresponded to 3.00 p.m. Light cabinet adjusted mice were allowed to acclimatize to the change in light/dark schedule for at least 2 weeks before immunization. MOG + CFA emulsion was made up 3 weeks prior to immunization to ensure that each group of mice received a standard immunization. Draining LN cells were isolated on d 14 of EAE and re-stimulated ex vivo with medium or with $MOG_{35-55}$ antigen (25 or 50 μg/ml) for 3 d and the final 4 h in the presence of PMA (Sigma-Aldrich, 10 ng/ml), ionomycin (Sigma-Aldrich, 1 μg/ml) and brefeldin A (10 μg/ml). Supernatants were removed and tested by ELISA for IL-17 (R&D Systems M1700), GM-CSF (R&D Systems MGM00), IL-10 (R&D Systems DY-417-05) and IFN-γ (BD Biosystems, capture antibody 551216, detection antibody 554410) while the cells were examined by FACS analysis. For peritoneal exudate cell (PEC) analysis, cells were removed from mice 3d after immunization of naive female C57BL/6 mice with MOG35–55 + CFA + PT at ZT6 or ZT18. EAE disease scores was recorded as follows: grade 0, normal; grade 1, limp tail; grade 2, wobbly gait; grade 3, hind limb weakness; grade 4, hind limb paralysis; and grade 5, tetraparalysis/death.

**Isolation of mononuclear cells from CNS**. Mice were killed and perfused with phosphate-buffered saline, and their spinal cords and brains isolated. Brain and spinal cord mononuclear cells were then purified by first disrupting the spinal cord using a tissue lyser then using density gradient centrifugation in Percoll or using a Multi tissue dissociation kit (Miltenyi 130-110-201) in combination with a gentleMACS Dissociator (Miltenyi) followed by debris removal step (Miltenyi). Isolated cells were stimulated for 4 h with PMA, ionomycin in the presence of brefeldin A (IL-17, IFN-γ and GM-CSF production) or for 2 h with brefeldin A (IL-1β production, R&D Systems DY-401-05) and then surface and intracellularly stained and analyzed by FACS.

**Flow cytometry**. LN, PEC, expanded BM and infiltrating mononuclear cells isolated from spinal cords were washed before being incubated with a live/dead stain. Cells were then incubated with an Fcγ block (BD) and surface stained with antibodies specific for CD11b (clone M1/70, 1/100, eBioscience), CD11c (clone N418, 1/100, Biolegend), Ly6C (clone AL-21, 1/200, BD Biosciences), Ly6G (clone RB6-8C5, 1/200, eBioscience), CCR2 (clone SA203G, 1/100, Biolegend), MHCII (clone M5/117.15.2, 1/200, eBioscience), PD-L1 (clone MIH5, 1/100, eBioscience), PD-L2 (clone TY25, 1/100, Biolegend), CD45 (clone 30-F11, 1/200, Biolegend), CD3 (clone 17A2, 1/200, eBioscience), CD4 (clone RM4-4, 1/100, eBioscience) and CD8 (clone 53¯6.7, 1/100, eBioscience). For intracellularly stained samples cells were then washed, fixed and permeabilized using 2% PFA (Pierce) and 0.5% saponin (Sigma-Aldrich) containing antibodies specific for IL-17 (clone TC11-18H10-1, 1/100, eBioscience), IFN-γ (clone XMG1.2, 1/100, eBioscience), or IL-1β (clone NJTEN3, 1/50, eBioscience). Cells were analysed using a LSRFortessa flow cytometer (BD) and the data was analysed with FloJo software. Analysis of the stained populations was performed by gating on single, live cells. Gating strategy is illustrated in Supplementary Fig. 8.

$CD11c^+MHCII^+$ and $CD11b^+Ly6C^{hi}$ populations of immune cells were FACS-sorted from single cells suspensions of spleens from individual mice ($n = 5$/time point), using a FACSAria Fusion High Performance Cell Sorter (BD). Cells were sorted on ice directly into RNA Later (Ambion), re-suspended in TRIzol (Invitrogen) and frozen at −20 °C.

Reverse transcription-PCR. RNA was extracted from brain, spinal cord, LN, expanded BM and FACS-sorted immune populations using the chloroform/isopropanol method and was reverse transcribed into cDNA using a High Capacity cDNA Reverse Transcription Kit (cat. no. 4368814, Applied Biosystems). RT-PCR was performed using commercially available Bmal1 (Mm00500226), Reverbα (Mm00520708), Per2 (Mm00478113), Clock (Mm00455950), Il1b (Mm00434228), Ifng (Mm01168134), Csf2 (Mm01290062), Il17α (Mm00439618) and Ccl2 (Mm00441242) primers (ABI). RT-PCR was performed on an ABI PRISM7500 Sequence Detection System (Applied Biosystems). The amount of each gene was determined by normalization to 18 S rRNA Mm04277571 or GAPDH (Mm99999915) internal controls.

**Expansion of BM cells with GM-CSF or M-CSF**. BM was obtained from the tibia and femurs of naive $Bmal1^{Mye+/+}$ or $Bmal1^{Mye-/-}$ mice. Briefly, BM was flushed

from the bones using a 23 G needle and a single cell suspension was obtained by passing the BM through a 19 G needle. Red blood cells were lysed and immature monocytes were cultured in RPMI 1640 medium supplemented with either 20 ng/ml GM-CSF in the form of supernatant from the J558 plasmacytoma cells or with M-CSF (100 ng/ml) in the form of supernatant from the L929 cell line. Cells were fed with GM-CSF or M-CSF containing media on d 3 and were harvested on d 6. For GM-CSF-expanded cells loosely adherent cells were harvested while for M-CSF-expanded cells flasks were scraped to isolate the adherent cells. Prior to stimulation BM isolated cells were allowed to rest for at least 5 h. Isolated cells were stimulated with MTB (100 μg/ml) or with medium alone. Supernatants were removed after either 4 h or 24 h and tested by ELISA for cytokine production or isolated and washed for FACS analysis. For the generation of antigen-specific responses GM-CSF or M-CSF-expanded BM cells were incubated for 4 h with MTB and $MOG_{35-55}$ antigen (25 μg/ml). MOG-specific CD4 T cells were generated by immunizing mice with MOG + CFA, without PT, for 7 d. CD4 T cells were isolated by MACS sorting (Miltenyi) from the draining LN of immunized mice and co-cultured with BM derived cells at a ratio of 1 BM cell: 10 CD4 T cells. After 72 h co-stimulation supernatants were removed and tested by ELISA for IL-17, IFN-γ, GM-CSF and IL-1β production.

**Statistics**. All data was analysed using Prism 5 (GraphPad Software). Unpaired $t$ tests, Mann–Whitney, one way ANOVA or area under the curve tests were performed on datasets. Error bars represent standard error of the mean (SEM). Details provided in figure legends.

**Data availability**. All relevant data are available, on request, from the corresponding authors.

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

## Acknowledgements

We would like to thank Barry Moran for his help and expertize in FACS sorting of immune cell populations for the time-of-day experiments. This work was supported by Science Foundation Ireland (SFI) Principle Investigator Research Grant 11/PI/1036 and an SFI infrastructure grant (12/RI/2340) to K.H.G.M and the SFI Starting Investigator Research Grant 13/SIRG/2130 to A.M.C.

## Author contributions

C.E.S., C.M.F. and A.M.C. designed the experiments. K.H.G.M. and L.A.J.O'N. provided critical expertize in the design of the project. C.E.S performed the majority of the experiments. C.M.F., M.R. and J.D. performed the time-of-day EAE experiments. M.R. and J.D. performed adoptive transfer experiments. C.M.F. assisted with statistical analysis. J.O.E. assisted with the time-of-day and qPCR experiments, Z.Z. carried out some of the M-CSF-expanded bone marrow experiments. C.E.S. and A.M.C. wrote and revised the manuscript. C.M.F., L.A.J.O'N. and K.H.G.M. provided critical feedback on the manuscript.

## Additional information

**Competing interests:** The authors declare no competing financial interests.

