## [Peer Review File · Nature Communications]

Reviewers' comments:

Reviewer #1 (Circadian-immune crosstalk) (Remarks to the Author):

The manuscript describes the link between BMAL1 in myeloid cells and increased cellular responses in a model of EAE. It shows that absence of BMAL1 is linked to a more severe disease and a stronger inflammatory response. Although the work is important and clearly written, there are some points of criticism.

Major

In several instances, the authors wrote "increased", "loss of", "increase", "enhanced" and similar ones without giving significance levels. E.g., in Figures 1d (PD-L2), 7a, suppl. Figures 1d, 1e, 1f, 3d, 4, 5b, and 5c. A clear statement needs a clear statistic test.

Results, section "Loss of Bmal1 and Reverba correlates with an increased inflammatory environment", Clock and Per2 are not regulated: These two factors should be regulated because they are in the feedback loops of the BMAL1 and reverba. This needs to be discussed.

In the studies on "Effect of time-of-day immunisation on severity of EAE" the link to the BMAL1 story is missing. Any good idea to overcome this?

There is a general problem with the expression "All data from one experiment representative of three". If you make three experiments on a certain number of mice but you present only one experiment, the question comes up: Why have all the other mice suffered from experimentation? The ethical issue is surfacing. In addition, it is highly questionable why only showing one experiment. There are good possibilities to combine experiments which also increases the number of animals or samples. This needs to be done.

Another point is the demonstration of data. Giving the data as in figure 3b is good because it shows the individual samples in an immediate way. The reader can judge the story immediately. However, in many graphs the authors show the outdated "dynamite bunker plot." Usually, the data in these types of experiments are not normally distributed, they are skewed. This can be nicely seen in figure 4c, where two outliers make the show. Using non-parametric tests (Mann-Whitney for two different groups), box plots with superimposed symbols for each individual sample, and the median instead of the mean overcomes the problem. The authors should present the data of all experiments in this way.

Sometimes they use ANOVA and T-tests upon $n=3$ or $n=4$. The numbers are too low to use these test in a meaningful way. Again, combination of data of all experiments would overcome the problem.

In the figure legends, sometimes the number of individual samples is missing. Check this carefully.

In figure 3a, when have the statistics been performed (day?)? What about the other time points? When it is given like in figure 7, one needs to mention the problem of "multiple comparisons." In these disease score over time graphs, one needs to compare groups with other test that include all data and all groups in one single test. Seek the advice of a statistician.

Sometimes, eg. In Fig. 7b/IL-6 or Fig.2/GM-CSF, the authors have a value of zero that is compared to a mean. If one compares a zero with something else, a t-test is not sufficient. It must be a one sample test. In the case of non-parametric data, it is a Wilcoxon one sample test, which should be used here. Seek the advice of a statistician.

Suppl. Figure 5a: One outlier makes the show. This is not convincing.

Minor

Introduction, 4th paragraph: The authors have written "Presentation of antigen by the TCR, through regulation...". This is logically wrong because the TCR does not present the antigen.

Results: Explain MTB and PT on first use.

Results, section "Loss of Bmal1 from GM-CSF-expanded BM augments their ability to prime antigen specific Th1 and Th17 cells", sentence "...when the BM was derived from...": Something is missing in the sentence making clear that it is a comparison.

Results, section "Bmal1 expression in myeloid lineage cell attenuates the development of EAE", sentence "...indicating that CD11b+Ly6ChiCCR2+ cells are crucial for the development of EAE.": This sounds too causal here in this paper. The causality has not been demonstrated.

Same section in Results, Sentence "..., where Bmal1 was not depleted, ...": The "not" in the sentence should be removed, isn't it?

Results, section "Effect of time-of-day immunisation on severity of EAE", sentence "This correlated with diurnal oscillations in the numbers of CD11b+Ly6Chi and CD11c+MHCII+ cells in the spleen peaking at ZT6...": This is only true for CD11b+Ly6Chi and not for CD11c+MHCII+. Needs to be re-written.

Results, section "Effect of time-of-day immunisation on severity of EAE", sentence "...and this trend was also observed in the Ly6Chi population...": This needs to be re-written as "... and this phenomenon was also observed in the Ly6Chi population in form of a trend..."

Discussion, paragraph starting with "Our findings suggest that there is dysregulation of clock genes..." The authors wrote "Per or Cry" but the exact gene whether Per1, Per2, Per 3, Cry1, Cry 2 should be mentioned. There is lot of complexity.

Discussion: The authors correctly wrote that dysregulation of clock genes added to inflammation and autoimmune disease but also the opposite statement is true. Autoimmune disease changes the expression and rhythmicity of clock genes (e.g., Arthritis Res Ther. 2012 May 23;14(3):R122. doi: 10.1186/ar3852.).

Discussion: Last sentence: The time-of-day drug targeting is already successful in patients with rheumatoid arthritis. See Buttgerit et al. Lancet. 2008 Jan 19;371(9608):205-14.

Reference 39: It might be better to use the very first article showing this phenomenon: Psychosom Med. 2003 Sep-Oct;65(5):831-5. and J Immunol. 2011 Jul 1;187(1):283-90.

Figure legend to figure 1: The dose of M-CSF is given as 20% v/v. This does not mean anything to the reader. It should be ng/ml.

Figure legend to figure 7: Explain PEC. What does "+" and "++" mean?

Figure 1: Replace the panel indicator "s" with "d"

All figures: Indicate the level of significance for all comparisons either by giving the sig. p-value or "n.s." This makes it more clear.

Figure 4d, The authors need to provide a statistic test to make the clear statement given in the text.

Figure suppl. 3d: Why is mouse #1 and #2 so completely different compared to the rest?

Reviewer #2 (Multiple sclerosis/neuro-immune crosstalk) (Remarks to the Author):

This paper studies the function of a gene involved in circadian rhythmicity, *Bmal1*, in myeloid cells during brain autoimmune responses, a tantalizing, and timely issue. Focusing on macrophages, the work extends and complements recent studies of circadian changes in autoimmune T lymphocytes. Much of the work has been done with proven expertise, but a number of shortcomings preclude publication at this point.

An important question that the authors should address is whether the observed rhythmicity in EAE can be ablated by cell-autonomous targeting of the myeloid-cell clock. Further, although the paper focuses on myeloid cells, it remains unclear which immune cell type (innate or adaptive) is the major driver for the observed enhanced inflammatory CNS phenotype: Is it Ly6C⁺ inflammatory monocytes that due to loss of *Bmal1* are more pro-inflammatory – or is the effect secondary, due to the infiltration of *Bmal1*-sufficient T cells that nevertheless are more inflammatory due to prior interactions with pro-inflammatory myeloid cells?

Both GM-CSF expanded CD11c⁺, and M-CSF expanded CD11b⁺ cells are depleted of *Bmal1*, but "intriguingly", this is not the case in ex vivo splenic CD11c⁺ cells. Are there any phagocytes in vivo that replicate the behavior of cultured cells? This warrants a more detailed study, and a detailed definition of the "Ly2 lineage". How representative of natural phagocytes are culture derived lookalikes? A description of the transgenics is missing.

MTB (please define acronym) activation enhances IL-1b and IL-23, but hardly IL-6 (Fig.1b).

Culture induced KO cells show stronger response to MTB than WT cells, and present MOG better to CD4 T cells (from WT donors?). Why was MTB and MOG combined, was pure MOG presentation also enhanced? Remarkably, CD4⁺ T cells from KO mice exhibit stronger response to MTB/MOG than WT T cells. This impinges on interpretation of EAE experiments.

*Bmal1*KO mice challenged with MOG/CFA develop enhanced CNS inflammation, but is this due to the intrinsic hyperreactivity of *Bmal1* sufficient CD4 T cells, are to accessory cells (APC)? Which of the myeloid cells are KO, and which one are not? Also, please give your EAE score definition.

In d3 *Bmal1*KO draining LN CD11b are increased, but what about CD11c?

Pro-inflammatory factors are enhanced in KO mice, but the difference is surprisingly low. This may well be due to the study of brain tissues, which in the C57BL model are less affected than spinal cord. The latter should be included. Also *Bmal1* expression should be checked in cell types.

Timing of CNS sampling is confusing. Why was the cytokine response in the LNs analyzed at day 14, a time, when the priming phase should be long over with most of the effector cells en route to the CNS? In stark contrast, responses in the CNS (b-e) were analyzed on day 10, well before clinical disease.

Infiltrates of PTlow EAE basically reflect clinical ratings (grade <1 in *Bmal1*wt vs 2 in *Bmal1*KO), but is this due to *Bmal1*KO accessory cells, or to *Bmal1*WT CD4 effector cells(Fig.4)?

Fig.6 compares MOG/CFA/PT EAE mice with naïve ones. Including CFA/PT-only mice (severe systemic inflammation!) would distinguish systemic from EAE/CNS effects!).

In Fig.7, peritoneal exudates are used for myeloid cell analysis. Does exudate induction (induction is not described in M&M) affect the responses?

The diurnal effect of EAE immunization is truly amazing. It is known that immune cells vary in their momentary inflammatory responsiveness through the day (Fig.7a), but how can this circadian variability affect the long-term periods of stimulation and EAE build-up, which both last over days, if not weeks? Is the very initiation phase of these responses all decisive? How would change of day/night rhythmicity during the subsequent days affect the result? Perhaps these questions go beyond the scope of the present work. An important question that the authors should address is whether the observed rhythmicity in EAE can be ablated by cell-autonomous targeting of the myeloid-cell clock.

Reviewer #3 (Circadian- arthritis) (Remarks to the Author):

The authors provide a further evidence regarding the concept that the immune responses follow a circadian rhythm.

The focus on peripheral circadian clocks mechanisms.

However, the authors must first describe shortly the function of the CNS biological clock that is partially responsible for the circadian rhythms involving the peripheral molecular clocks.

The authors must introduce the concept the the neuro and adrenal hormonal control of the circadian activation of the immune responses as mediated by the nocturnal hormones like melatonin and cortisol.

To help to find the sources we suggest some references.

As matter of fact the authors show to be aware of the "solar" influence (darkness and light influence the circadian rhythms of the immune responses, as well as conditions as shift night workers or long flights) that they also mention.

Air Travel, Circadian Rhythms/Hormones, and Autoimmunity.

Torres-Ruiz J, Sulli A, Cutolo M, Shoenfeld Y.

Clin Rev Allergy Immunol. 2017 Feb 27

Glucocorticoids and chronotherapy in rheumatoid arthritis.

Cutolo M.

RMD Open. 2016 Mar 18;2(1):e000203. doi:

A Role of the neuroendocrine network on the immune responses (here including the cells studied by the authors) must be discussed and some sentences reported:

Role of neuroendocrine and neuroimmune mechanisms in chronic inflammatory rheumatic diseases-- the 10-year update.

Straub RH, Bijlsma JW, Masi A, Cutolo M.

Semin Arthritis Rheum. 2013 Dec;43(3):392-404.

The results observed by the authors should be also briefly discussed on the light of epigenetic mechanisms (modulation of the genes), that can explain the links between the neuro hormones of the CNS (central clock) and other steroids and the molecular responses of the peripheral cells with their circadian rhythms.

Cardinal Epigenetic Role of non-coding Regulatory RNAs in Circadian Rhythm.

Bhadra U, Patra P, Pal-Bhadra M.

Mol Neurobiol. 2017 May 17. doi: 10.1007/s12035-017-0573-8.

Epigenetic and Posttranslational Modifications in Light Signal Transduction and the Circadian Clock in *Neurospora crassa*.

Proietto M, Bianchi MM, Ballario P, Brenna A.

Int J Mol Sci. 2015 Jul 7;16(7):15347-83.

Therefore, several autoimmune disorders , and not only multiple sclerosis, present altered circadian rhythms such as rheumatoid arthritis (for example).

The authors should mention.

Rheumatoid arthritis: circadian and circannual rhythms in RA.

Cutolo M.

Nat Rev Rheumatol. 2011 Aug 2;7(9):500-2

Chronobiology and the treatment of rheumatoid arthritis.

Cutolo M.

Curr Opin Rheumatol. 2012 May;24(3):312-

Point by Point Response to the Reviewer's Comments

Reviewer #1

The manuscript describes the link between BMAL1 in myeloid cells and increased cellular responses in a model of EAE. It shows that absence of BMAL1 is linked to a more severe disease and a stronger inflammatory response. Although the work is important and clearly written, there are some points of criticism.

Authors Response: We thank the Reviewer for the complementary comments on our manuscript.

Major Points

In several instances, the authors wrote “increased”, “loss of”, “increase”, “enhanced” and similar ones without giving significance levels. E.g., in Figures 1d (PD-L2), 7a, suppl. Figures 1d, 1e, 1f, 3d, 4, 5b, and 5c. A clear statement needs a clear statistic test.

Authors Response: We have amended the text to clarify results that showed a clear statistical difference. Results that did not meet significance are now stated as non-significant (NS) or have been removed.

Loss of Bmal1 and Reverba correlates with an increased inflammatory environment”, Clock and Per2 are not regulated: These two factors should be regulated because they are in the feedback loops of the BMAL1 and reverba.

Authors Response: The Reviewer is correct in stating that under normal conditions *Bmal1* expression should positively impact on *Per2*. However, in an inflammatory environment, such as during experimental autoimmune encephalomyelitis (EAE), it appears that the feedback loops of the clock are disrupted. Lipopolysaccharide (LPS) stimulation of macrophages also disrupts clock feedback loops. We have reported a reduction of *Bmal1* and *Rev-Erba*, but an increase in *Per2* mRNA following LPS stimulation of macrophages (Curtis et al. PNAS, 2015 Jun 9;112(23):7231-6). We have clarified this point in the revised manuscript (Lines 210-213).

“In the studies on “Effect of time-of-day immunisation on severity of EAE” the link to the BMAL1 story is missing. Any good idea to overcome this?”

Authors Response: We now include additional text where we discuss the link between time-of-day immunization and *Bmal1*. Our time-of-day experiments show that EAE is under circadian influence, much like what has been observed in previous studies with bacterial, viral and LPS induced responses. Consistent with a review by Man et al (Science 2016 Nov 25;354(6315):999-1003), who concluded that as a whole a functional clock within a cell is anti-inflammatory, an animal lacking myeloid *Bmal1*, and therefore lack a functional clock in these cells, the pro-inflammatory responses are heightened. We found that the course of disease was heightened when EAE was induced at ZT6, a time when BMAL1 expression is high. Therefore, it is more likely that having a functional clock, rather than BMAL1 itself, confers the anti-inflammatory and protective effect in EAE. These points have now been included in the discussion section (Lines 342-346).

There is a general problem with the expression “All data from one experiment representative of three”. If you make three experiments on a certain number of mice but you present only one experiment, the question comes up: Why have all the other mice suffered from experimentation? The ethical issue is surfacing. In addition, it is highly questionable why only showing one experiment. There are good possibilities to combine experiments which also increases the number of animals or samples. This needs to be done.”

Authors Response: Where possible we have combined data from a number of experiments, and modified the text and figure legends accordingly. However, in certain situations, especially for EAE experiments, this is sometimes not possible because of the fact that the peak scores for the WT mice can vary between experiments. However, this does not affect the comparison between wildtype and knockout within individual experiments.

Another point is the demonstration of data. Giving the data as in figure 3b is good because it shows the individual samples in an immediate way. The reader can judge the story immediately. However, in many graphs the authors show the outdated “dynamite bunker plot.” Usually, the data in these types of experiments are not normally distributed, they are skewed. This can be nicely seen in figure 4c, where two outliers make the show. Using non-parametric tests (Mann-Whitney for two different groups), box plots with superimposed symbols for each individual sample, and the median instead of the mean overcomes the problem. The authors should present the data of all experiments in this way.

Authors Response: We have now presented (where possible) the data as scatter graphs and performed Mann Whitney tests for statistical comparisons.

Sometimes they use ANOVA and T-tests upon $n=3$ or $n=4$. The numbers are too low to use these test in a meaningful way. Again, combination of data of all experiments would overcome the problem.

Authors Response: While it is not ideal to use ANOVA or a T-tests for statistical analysis of data with small numbers of replicates, it is valid. Nevertheless, we have where possible combined data to increase the number of replicates.

In the figure legends, sometimes the number of individual samples is missing. Check this carefully.

Authors Response: We thank the Reviewer for pointing this out. As we have now changed the format of the figures to scatter plots the number of individual samples is clear, and we have also included this number in the figure legends.

In figure 3a, when have the statistics been performed (day?)? What about the other time points? When it is given like in figure 7, one needs to mention the problem of “multiple comparisons.” In these disease score over time graphs, one needs to compare groups with other test that include all data and all groups in one single test. Seek the advice of a statistician.

Authors Response: We had used multiple comparisons to analyze the data in Figure 3a. In all EAE experiments we have now used Kruskal-Wallis test on each day. For the ZT6/ZT18 experiments we have also employed the Mann-Whitney test of area under the curve.

“Sometimes, eg. In Fig. 7b/IL-6 or Fig.2/GM-CSF, the authors have a value of zero that is compared to a mean. If one compares a zero with something else, a t-test is not sufficient. It must be a one sample test. In the case of non-parametric data, it is a Wilcoxon one sample test, which should be used here. Seek the advice of a statistician.”

Authors Response: Based on the reviewers prior suggestion to pool experiments, we have now revised Figure 1b and Figure 2 substantially. The cytokines that were significantly different were IL-1 β , IL-23 and IFN- γ and on the recommendation of the reviewer we have used the Mann-Whitney test as the statistical test.

“Suppl. Figure 5a: One outlier makes the show. This is not convincing”.

Authors Response: We thank the Reviewer for the comment. The correlation between *Il1b* and *Csf2* does not significantly add to the manuscript and we have decided to remove that data. The key correlation is between *Bmal1* expression and EAE score (revised now to Fig 5c). Lower *Bmal1* expression is associated with more severe EAE and this result is in agreement with the experiments in *Bmal1*^{-/-} mice.

Minor Points

Introduction, 4th paragraph: The authors have written “Presentation of antigen by the TCR, through regulation...”. This is logically wrong because the TCR does not present the antigen.

Authors Response: This should of course have read ‘to’ not ‘by’ the TCR and has now been corrected. For further clarity this sentence has now been revised to ‘Regulation of the adaptor protein ZAP70, which controls antigen-induced T cell activation, is controlled in a circadian manner, leading to T cell proliferation responses that are dependent on time of day’ (Lines 81-83).

Results: Explain MTB and PT on first use.

Authors Response: We have now explained *MTB* (*Mycobacterium tuberculosis*) (Line 127) and PT (pertussis toxin); (Line 158) in the revised manuscript.

Results, section “Loss of Bmal1 from GM-CSF-expanded BM augments their ability to prime antigen specific Th1 and Th17 cells”, sentence “...when the BM was derived from...”: Something is missing in the sentence making clear that it is a comparison.

Authors Response: We have now amended this sentence to ‘Co-culture of GM-CSF-expanded BM cells with MOG-specific CD4 T cells, isolated from *Bmal1*^{Mye^{+/+}} mice 7 d after immunization with MOG+CFA, produced higher concentrations of IFN- γ when the BM was derived from *Bmal1*^{Mye^{-/-}} mice’. (Lines 138-140).

Results, section “Bmal1 expression in myeloid lineage cell attenuates the development of EAE”, sentence “...indicating that CD11b⁺Ly6ChiCCR2⁺ cells are crucial for the development of EAE.”: This sounds too causal here in this paper. The causality has not been demonstrated.

Authors Response: We have revised the sentence to: ‘We found that the Ly6C^{hi}CCR2⁺ cells, which infiltrate the brain during EAE, are entirely CD11b⁺ (revised now to Supplemental Fig. 5a, b), indicating that CD11b⁺Ly6C^{hi}CCR2⁺ cells may be important for the development of EAE (Lines 176- 178).

Same section in Results, Sentence “..., where Bmall was not depleted, ...”: The “not” in the sentence should be removed, isn’t it?

Authors Response: This error has been corrected.

Results, section “Effect of time-of-day immunisation on severity of EAE”, sentence “This correlated with diurnal oscillations in the numbers of CD11b+Ly6Chi and CD11c+MHCII+ cells in the spleen peaking at ZT6...”: This is only true for CD11b+Ly6Chi and not for CD11c+MHCII+. Needs to be re-written.

Authors Response: The Reviewer is correct and we have amended the text to ‘This correlated with diurnal oscillations in the numbers of CD11b⁺Ly6C^{hi} cells, but not CD11c⁺MHCII⁺ cells, in the spleen peaking at ZT6 (now presented in Supplemental Fig. 7a)’ (Lines 228 - 230).

Results, section “Effect of time-of-day immunisation on severity of EAE”, sentence “...and this trend was also observed in the Ly6Chi population...”: This needs to be re-written as “... and this phenomenon was also observed in the Ly6Chi population in form of a trend...”

Authors Response: This sentence has now been amended in line with the Reviewer’s suggestion (Line 252).

Discussion, paragraph starting with “Our findings suggest that there is dysregulation of clock genes...” The authors wrote “Per or Cry” but the exact gene whether Per1, Per2, Per 3, Cry1, Cry 2 should be mentioned. There is lot of complexity.

Authors Response: We have amended the text to read ‘Indeed we show the selective reduction in mRNA expression of *Bmall* and *Reverba*, but not *Period2* or *Cryptochrome2*, in the CNS of mice with EAE’ (Lines 323- 324).

Discussion: The authors correctly wrote that dysregulation of clock genes added to inflammation and autoimmune disease but also the opposite statement is true. Autoimmune disease changes the expression and rhythmicity of clock genes (e.g., Arthritis Res Ther. 2012 May 23;14(3):R122. doi: 10.1186/ar3852.).

Authors Response: We thank the Reviewer for this helpful suggestion. We have now added this point to the discussion: ‘Autoimmune and degenerative/inflammatory disease such as rheumatoid arthritis and osteoarthritis can impact directly on molecular clock expression, thus potentially driving to a vicious cycle of inflammation’ (REF: Arthritis Res Ther. 2012 May 23;14(3):R122. doi: 10.1186/ar3852.). (Lines 326- 329).

Discussion: Last sentence: The time-of-day drug targeting is already successful in patients with rheumatoid arthritis. See Buttgeret et al. Lancet. 2008 Jan 19;371(9608):205-14.

Authors Response: We have now incorporated this and the Buttgeret reference in the discussion section after the text “ Indeed, therapeutic strategies which incorporate alignment to circadian rhythms, such as alignment of prednisolone administration to the endogenous circadian rhythm of cortisol, has proved efficacious in treating rheumatoid arthritis”. (Lines 331 -335).

Reference 39: It might be better to use the very first article showing this phenomenon: Psychosom Med. 2003 Sep-Oct;65(5):831-5. and J Immunol. 2011 Jul 1;187(1):283-90.

Authors Response: We thank the Reviewer for this suggestion and we have revised the second last paragraph of the discussion and included the two references suggested by the Reviewer (Lines 334 - 335).

Figure legend to figure 1: The dose of M-CSF is given as 20% v/v. This does not mean anything to the reader. It should be ng/ml.

Authors Response: The dose of M-CSF dose has now been given as 100 ng/ml (Line 422)

Figure legend to figure 7: Explain PEC. What does “+” and “++” mean?

Authors Response: PEC means peritoneal exudate cells; this has been spelled out in the revised manuscript (Line 247). The + and ++ referred to statistical significance but has now been amended to * and **.

Figure 1: Replace the panel indicator “s” with “d”

Authors Response: This typo has been corrected on the fully revised Figure 1.

All figures: Indicate the level of significance for all comparisons either by giving the sig. p-value or “n.s.” This makes it more clear.

Authors Response: These changes have been made across all figures.

Figure 4d, The authors need to provide a statistic test to make the clear statement given in the text.

Authors Response: Figure 4d has now been revised to Figure 3d, and we have now provided a dot plot of the data and statistical analysis.

Figure suppl. 3d: Why is mouse #1 and #2 so completely different compared to the rest?

Authors Response: In the EAE model, mice normally display clinical signs on different days. In the *Bmal1*^{+/+} group mouse 1 and 2 had clinical signs of EAE and therefore displayed proinflammatory responses in the brain as expected, the other 4 *Bmal1*^{+/+} mice remained well and so did not have brain infiltration. However, each of the 6 *Bmal1*^{-/-} KO all had clinical signs of EAE and therefore all had pro-inflammatory infiltration.

Reviewer #2

This paper studies the function of a gene involved in circadian rhythmicity, Bmal1, in myeloid cells during brain autoimmune responses, a tantalizing, and timely issue. Focusing on macrophages, the work extends and complements recent studies of circadian changes in autoimmune T lymphocytes.

Much of the work has been done with proven expertise, but a number of shortcomings preclude publication at this point.

An important question that the authors should address is whether the observed rhythmicity in EAE can be ablated by cell-autonomous targeting of the myeloid-cell clock

Authors Response: We have now conducted experiments where myeloid *Bmal1*^{-/-} and control mice were immunized at ZT6 and ZT18 under the same conditions as the experiment shown in Fig. 7B. Strikingly, we found that the rhythmicity observed

between ZT6 and ZT18 was lost in mice lacking *Bmal1* in myeloid cells. This data suggests that indeed the myeloid clock determines the rhythmicity in EAE observed in the Wild-type mice. This new data has been included in Figure 6 and discussed Line 241-243, of the revised manuscript. We thank the Reviewer for this suggestion and we believe this adds significantly to our findings regarding the importance of the circadian system in autoimmune disease.

Is it Ly6C⁺ inflammatory monocytes that due to loss of Bmal1 are more pro-inflammatory – or is the effect secondary, due to the infiltration of Bmal1-sufficient T cells that nevertheless are more inflammatory due to prior interactions with pro-inflammatory myeloid cells?

Authors Response: We thank the Reviewer for posing this question. Our data suggest that myeloid cells lacking *Bmal1* are producing more IL-1 β and thereby enhance Th17 responses which are pathogenic in EAE. In support of this conclusion, we observed increases in IL-1 β in the brain (Fig. 2b) and spinal cord (Fig. 3c) of *Bmal1*^{Mye^{-/-}} mice on day 10 of EAE. Furthermore, when GM-CSF-amplified bone marrow derived *Bmal1*^{Mye^{+/+}} cells were stimulated with MOG and MTB prior to co-culture with T cells derived from MOG+CFA immunized mice, we observe an increase in IL-17 and IFN- γ when the T-cells were derived from *Bmal1*^{Mye^{-/-}} compared with *Bmal1*^{Mye^{+/+}} mice (Supplemental Fig. 4). Our interpretation of these findings is that the T-cells are more pro-inflammatory due to interaction with the *Bmal1* deficient myeloid cells in the periphery, in part because they produce more IL-1 β . However, we can not rule out a role for *Bmal1*^{Mye^{-/-}} cells migrating into the CNS. We have further interrogated this by using the adoptive T cell transfer model of EAE. Spleen and LN cells from *Bmal1*^{Mye^{+/+}} and *Bmal1*^{Mye^{-/-}} mice immunized with MOG and CFA were expanded in vitro for 3 days with MOG, IL-1 β and IL-23 (our standard protocol for expanding MOG-specific Th17 cells). Purified CD3⁺ cells (purity < 97%) from these cultures were injected i.p. into WT naïve recipient mice. The mice developed EAE by day 8 post injection and there was no difference in the clinical scores when the T cells were derived from *Bmal1*^{Mye^{+/+}} or *Bmal1*^{Mye^{-/-}} (see A panel below). Furthermore, we did not observe an increase in either IL-17 or IFN- γ in the cultures (B panel). We believe that the IL-1 β that we added to the culture overcame the effects observed in the experiments described in Supplemental Figure 4. This provided further evidence that the increase in myeloid IL-1 β is a major factor in promoting enhanced disease in mice lacking myeloid *Bmal1*.

A

B

Are there any phagocytes in vivo that replicate the behavior of cultured cells? This warrants a more detailed study, and a detailed definition of the “Lyz2 lineage”. How representative of natural phagocytes are culture derived lookalikes? A description of the transgenics is missing.

Authors Response: While it can be difficult to extrapolate from cultured cells to the equivalent cells in vivo. It is accepted protocol to use in vitro expanded cells as a surrogate of the cells in vivo. We have included a detailed definition of the Lyz2 lineage (Lines 367) and a description of the transgenics (Lines 365) in the section of animals in Materials and Methods.

MTB (please define acronym) activation enhances IL-1b and IL-23, but hardly IL-6 (Fig.1b).

Authors Response: The MTB abbreviation has now been explained (Line 127). We have now pooled a number of experiments for Figure 1 and IL-6 has not been included in this new analysis.

Culture induced KO cells show stronger response to MTB than WT cells, and present MOG better to CD4 T cells (from WT donors?). Why was MTB and MOG combined, was pure MOG presentation also enhanced? Remarkably, CD4+ T cells from KO mice exhibit stronger response to MTB/MOG than WT T cells. This impinges on interpretation of EAE experiments.

Authors Response: MOG-specific memory Th1 and Th17 cells from mice immunized with MOG and CFA respond to antigen (MOG), but also require a third signal in the form of T cell polarizing cytokines, IL-12 for Th1 and IL-1 and IL-23 for Th17 cells. We have previously shown that MTB can induce IL-12, IL-1 and IL-23 from macrophages and dendritic cells (Sutton *et al.* J. Exp. Med. 203(7):1685-91 (2006) Therefore, MTB was added to the cultures to promote the T cell polarizing cytokines. Our previous studies had also revealed that stimulation with MOG alone induces little or no T cell cytokine production. We agree that the observations that the data showing that CD4⁺ T cells from KO mice exhibit stronger response to MTB/MOG than WT T cells is an interesting finding. This is consistent with the EAE data and the overall conclusions that T-cells are more pro-inflammatory due to interaction with the Bmal1 deficient myeloid cells that produce more IL-1 β .

Bmal1KO mice challenged with MOG/CFA develop enhanced CNS inflammation, but is this due to the intrinsic hyperreactivity of Bmal1 sufficient CD4 T cells, are to accessory cells (APC)? Which of the myeloid cells are KO, and which one are not? Also, please give your EAE score definition.

Authors Response: The question of whether the enhanced CNS inflammation is due to the myeloid cells or Bmal1 sufficient T cells has been discussed above.

In terms of which of the myeloid cells are KO and which are not, our studies (Curtis *et al.* PNAS, 2015 Jun 9;112(23):7231-6) as well as those of others (Nguyen *et al.* Science, 2013, Vol. 341, 1843), have shown that the monocytes macrophages in the knockout mice are devoid of *Bmal1*. We have unpublished data showing that in neutrophils there is a partial reduction of *Bmal1* expression in the *Bmal1*^{Mye^{-/-}} mice. *Bmal1* expression is very low in eosinophils in WT cells with a marginal reduction in cells from *Bmal1*^{Mye^{-/-}} mice. We have now included the EAE score definition in the Materials and Methods section (Line 392-394).

In d3 Bmal1KO draining LN CD11b are increased, but what about CD11c?

Authors Response: Although there was a trend for higher CD11c cells in the d3 Bmal1 KO draining LN, it was not significant.

Pro-inflammatory factors are enhanced in KO mice, but the difference is surprisingly low. This may well be due to the study of brain tissues, which in the C57BL model are less affected than spinal cord. The latter should be included. Also Bmal expression should be checked in cell types.

Authors Response: The enhancement of *Il1 β* , *Ccl2* and *Ifn- γ* in KO mice is significant at $P < 0.01$ (now revised to Fig 2b). We did not perform RT-PCR on the spinal cord, however, we did perform FACS analysis (now revised to Fig. 3a-c) and we clearly observe increased number and percentage of CD11b⁺Ly6C^{hi} and a very high frequency of IL-1 β -secreting CD11b⁺Ly6C^{hi} cells, supporting our hypothesis that loss of *Bmal1* from this cell type impacts on CNS inflammation. We have analysed the expression of *Bmal1*, this has been included in Supplemental Fig. 1B.

Timing of CNS sampling is confusing. Why was the cytokine response in the LNs analyzed at day 14, a time, when the priming phase should be long over with most of the effector cells en route to the CNS? In stark contrast, responses in the CNS (b-e) were analyzed on day 10, well before clinical disease.

Authors Response: For logistical reasons and to reduce the numbers of experiments on mice (for ethical reasons), we assessed LN on day 14, when we could also assess cell infiltration into the CNS. However, we do agree with the Reviewer that days 8-10 may show stronger responses in the LNs and later in the CNS. Nevertheless, responses are still robust in the LN on day 14 and the comparison here was between WT versus KO on the same day.

Infiltrates of PThow EAE basically reflect clinical ratings (grade <1 in Bmal1wt vs 2 in Bmal1KO), but is this due to Bmal1KO accessory cells, or to Bmal1WT CD4 effector cells(Fig.4)?

Authors Response: As discussed above, the T-cells are more pro-inflammatory due to interaction with the *Bmal1* deficient myeloid cells that are producing more IL-1 β .

Fig.6 compares MOG/CFA/PT EAE mice with naïve ones. Including CFA/PT-only mice (severe systemic inflammation!) would distinguish systemic from EAE/CNS effects!

Authors Response: We have now performed an additional experiment where WT mice we immunized CFA and PT (without MOG) and we did not observe the reduction in *Bmal1* and *Rev-Erb α* seen in mice immunized with MOG/CFA/PT. This suggests the observed effects on clock gene expression appears to be specific to EAE/CNS effects and not due to systemic inflammation. This data has now been included in Supplemental Fig. 6 and discussed Line 213-216.

In Fig.7, peritoneal exudates are used for myeloid cell analysis. Does exudate induction (induction is not described in M&M) affect the responses?

Authors Response: The peritoneal exudate cells were not specifically induced by any inflammatory agents injected into the peritoneal cavity, they were peritoneal cells recovered by lavage from mice 3 days after induction of EAE. This has been clarified further in the figure legend and Materials and Methods section (Lines 389-391).

The diurnal effect of EAE immunization is truly amazing. It is known that that immune cells vary in their momentary inflammatory responsiveness through the day (Fig.7a), but how can this circadian variability affect the long-term periods of stimulation and EAE build-up, which both last over days, if not weeks? Is the very initiation phase of these responses all decisive? How would change of day/night rhythmicity during the subsequent days affect the result? Perhaps these questions go beyond the scope of the present work.

Authors Response: We agree that the diurnal effect of EAE immunization is particularly striking and, as the Reviewer indicates, suggests that the initiation phase is decisive. We have further confirmed this by showing that this diurnal effect is dependent on myeloid Bmal1. It appears that at ZT6 the increased numbers of Cd11b⁺Ly6C^{hi} cells producing IL-1 β enhance the T cell response in the lymph node, which then leads to more inflammation in the brain. The suggestion to change the day/night rhythmicity during the subsequent days, while interesting might lead to additional confounding effects, such as on activity and feeding that may be difficult to interpret. As the reviewer indicates these questions go beyond the scope of the present work.

An important question that the authors should address is whether the observed rhythmicity in EAE can be ablated by cell-autonomous targeting of the myeloid-cell clock.

Authors Response: As discussed above, we have now performed this experiment and observe that the rhythmicity in EAE is indeed ablated by cell-autonomous targeting of the myeloid-cell clock, which significantly adds to our findings (Figure 6c).

Reviewer #3

The authors provide a further evidence regarding the concept that the immune responses follow a circadian rhythm.

The focus on peripheral circadian clocks mechanisms. However, the authors must first describe shortly the function of the CNS biological clock that is partially responsible for the circadian rhythms involving the peripheral molecular clocks. The authors must introduce the concept the the neuro and adrenal hormonal control of the circadian activation of the immune responses as mediated by the nocturnal hormones like melatonin and cortisol.

To help to find the sources we suggest some references. As matter of fact the authors show to be aware of the "solar" influence (darkness and light influence the circadian rhythms of the immune responses, as well as conditions as shift night workers or long flights) that they also mention. Air Travel, Circadian Rhythms/Hormones, and Autoimmunity. Torres-Ruiz J, Sulli A, Cutolo M, Shoenfeld Y. Clin Rev Allergy Immunol. 2017 Feb 27

Glucocorticoids and chronotherapy in rheumatoid arthritis. Cutolo M. RMD Open. 2016 Mar 18;2(1):e000203. doi:

Authors Response: We thank the Reviewer for pointing out our omission, and we have now included a full paragraph in the Introduction section on the central clock and hormonal control, and have incorporate the references above (Lines 44 to 54).

A Role of the neuroendocrine network on the immune responses (here including the cells studied by the authors) must be discussed and some sentences reported: Role of neuroendocrine and neuroimmune mechanisms in chronic inflammatory rheumatic diseases--the 10-year update. Straub RH, Bijlsma JW, Masi A, Cutolo M. Semin Arthritis Rheum. 2013 Dec;43(3):392-404.

Authors Response: We have included discussion points on the neuroendocrine network and the specific cells affected and incorporated the reference above (Lines 44-54).

The results observed by the authors should be also briefly discussed on the light of epigenetic mechanisms (modulation of the genes), that can explain the links between the neuro hormones of the CNS (central clock) and other steroids and the molecular responses of the peripheral cells with their circadian rhythms.

Cardinal Epigenetic Role of non-coding Regulatory RNAs in Circadian Rhythm. Bhadra U, Patra P, Pal-Bhadra M. Mol Neurobiol. 2017 May 17. doi: 10.1007/s12035-017-0573-8.

Epigenetic and Posttranslational Modifications in Light Signal Transduction and the Circadian Clock in Neurospora crassa. Proietto M, Bianchi MM, Ballario P, Brenna A. Int J Mol Sci. 2015 Jul 7;16(7):15347-83.

Authors Response: We thank the Reviewer for their helpful comments and have now included a reference in the discussion of the manuscript on the epigenetic changes that may be impacted by *Bmal1* deletion in myeloid cells, which is what is relevant to this paper (Lines 275-276).

Therefore, several autoimmune disorders , and not only multiple sclerosis, present altered circadian rhythms such as rheumatoid arthritis (for example). The authors should mention.

Rheumatoid arthritis: circadian and circannual rhythms in RA. Cutolo M. Nat Rev Rheumatol. 2011 Aug 2;7(9):500-2

Chronobiology and the treatment of rheumatoid arthritis. Cutolo M. Curr Opin Rheumatol. 2012 May;24(3):312-

Authors Response: The Reviewer is correct in highlighting that circadian rhythms can impact on a range of autoimmune disorders. We have now included additional discussion on the now well-established role of circadian rhythms on rheumatoid arthritis and the effective use of chronobiology in the treatment of rheumatoid arthritis (Lines 331-334).

REVIEWERS' COMMENTS:

Reviewer #1 (Remarks to the Author):

The paper has been substantially improved. There are no further points.

Reviewer #2 (Remarks to the Author):

I am satisfied by the authors' revision and recommend publication.

Reviewer #3 (Remarks to the Author):

The authors investigate on how the loss of the molecular clock in myeloid cells might exacerbate T-cell mediated CNS Autoimmune diseases.

There are some comments.

The authors must report more extensively about this key point Line 25:

PLEASE add:

".....add other autoimmune diseases such as rheumatoid arthritis"

The circadian clock regulates inflammatory arthritis.

Hand LE, Hopwood TW, Dickson SH, Walker AL, Loudon AS, Ray DW, Bechtold DA, Gibbs JE. FASEB J. 2016 Nov; 30(11): 3759-3770.

Circadian Clocks in the Immune System.

Labrecque N, Cermakian N.

J Biol Rhythms. 2015 Aug; 30(4): 277-90.

LINE 50:

It is not correct to refer to humans that melatonin is antiinflammatory. There are several evidences that the CNS circadian clock uses melatonin as started of the immune system activation at the beginning of the night /darkness).

The effect is then neutralise by the night time rise of cortisol (light).

Circadian Rhythm.

Please read and refer (report in references) the following studies:

Melatonin treatment does not improve rheumatoid arthritis.

Maestroni GJ, Otsa K, Cutolo M.

Br J Clin Pharmacol. 2008 May; 65(5): 797-8.

Circadian rhythms of nocturnal hormones in rheumatoid arthritis: translation from bench to bedside.

Cutolo M, Straub RH, Buttgereit F.

Ann Rheum Dis. 2008 Jul; 67(7): 905-8.

Circadian rhythms in rheumatology--a glucocorticoid perspective.
Spies CM, Straub RH, Cutolo M, Buttgereit F.
Arthritis Res Ther. 2014 Nov 13;16 Suppl 2:S3.

LINE 86

Same risk in has been found RA please or interference due to jet lag.

refer

Rheumatoid arthritis: should we shift the focus from "Treat to Target" to "Treat to Work?".
Almoallim H, Kamil A.
Clin Rheumatol. 2013 Mar;32(3):285-7.

line 269

Again it is suggested to report this reference:

Air Travel, Circadian Rhythms/Hormones, and Autoimmunity.
Torres-Ruiz J, Sulli A, Cutolo M, Shoenfeld Y.
Clin Rev Allergy Immunol. 2017 Aug;53(1):117-125.

Line 357

The authors should conclude that, as mentioned in ref 43, there are already examples of chronotherapy in autoimmune diseases.

Glucocorticoids and chronotherapy in rheumatoid arthritis.
Cutolo M.
RMD Open. 2016 Mar 18;2(1):e000203